# Conserved interfaces mediate multiple protein–protein interactions in a prokaryotic metabolon

Sanchari Bhattacharyya [1,2 ✉], Srivastav Ranganathan [1,3], Sourav Chowdhury [1,4], Bharat V Adkar [1,5], Mark Khrapko [1] & Eugene I Shakhnovich [1 ✉]

## Abstract

Enzymes in a pathway often form metabolons through weak protein–protein interactions (PPI) that localize and protect labile metabolites. Due to their transient nature, the structural architecture of these enzyme assemblies has largely remained elusive, limiting our abilities to re-engineer novel metabolic pathways. Here, we delineate a complete PPI map of 1225 interactions in the *E. coli* 1-carbon metabolism pathway using bimolecular fluorescence complementation that can capture transient interactions in vivo and show strong intra- and inter-pathway clusters within the folate and purine biosynthesis pathways. Scanning mutagenesis experiments along with AlphaFold predictions and metadynamics simulations reveal that most proteins use conserved "dedicated" interfaces distant from their active sites to interact with multiple partners. Diffusion-reaction simulations with shared interaction surfaces and realistic PPI networks reveal a dramatic speedup in metabolic pathway fluxes. Overall, this study sheds light on the fundamental features of metabolon biophysics and structural aspects of transient binary complexes.

**Keywords** Metabolon; Split YFP; Enzyme Flux; AlphaFold; Transient PPI
**Subject Categories** Metabolism; Structural Biology

## Introduction

Unlike purified enzymes in a test tube, enzymes inside cells often do not function in isolation. In many cases, they carry out their reaction through the formation of multi-enzyme assemblies. Metabolons—a term first coined by Paul Srere—are one such protein complex, which are transiently interacting supramolecular entities formed through weak interactions among sequential enzymes in a metabolic pathway that help to channel metabolites (Robinson and Srere, 1985; Srere, 1987). In higher eukaryotes, there has been evidence for such complexes that dynamically associate and dissociate in response to cellular stimuli, the most notable of them being purinosomes (An et al, 2008; Deng et al, 2012; French et al, 2016). It is generally postulated that the formation of a metabolon has several advantages, like metabolite channeling, increasing local concentration of metabolites, preventing competing pathways, and protection of unstable and cytotoxic intermediates (Sweetlove and Fernie, 2018; Zhang and Fernie, 2021). Despite this, the mechanistic understanding of how proteins in a metabolon interact has remained elusive. Moreover, due to the transient nature of the interactions, traditional methods of structural biology have failed to address these questions. These problems have largely impaired our abilities to design or redesign new enzyme pathways in metabolic engineering. Based on information from cross-linking studies and mass spectrometry, Wu and Minteer (2015) built a low-resolution structure of the tricarboxylic acid (TCA) cycle metabolon using constraint-based protein–protein docking. This study gave the first structural insight into a metabolon and showed evidence of electrostatic channeling upon association of proteins (Wu and Minteer, 2015).

More recently, it has been shown that even prokaryotic cells, which were once thought to be a bag of freely floating enzymes and metabolites, show evidence of metabolon formation and are therefore able to achieve a substantial degree of metabolic compartmentalization. In our previous work (Bhattacharyya et al, 2016), using immunoprecipitation experiments, we have shown that dihydrofolate reductase (DHFR), a central enzyme of the *E. coli* 1-carbon metabolism pathway, associates with multiple proteins in its functional vicinity via weak interactions, which in turn was also responsible for the overexpression toxicity of DHFR. Though precise evidence of substrate channeling was not established, a more recent work (Bhattacharyya et al, 2021) presented compelling circumstantial evidence that many of these interactions involve channeling of substrates, causing diffusion limitation of the substrates inside the cell. To obtain a deeper understanding of the breadth and nature of interactions in prokaryotic enzyme assemblies, in this work we established a

[1]Department of Chemistry and Chemical Biology, Harvard University, Cambridge, MA, USA. [2]Present address: Novartis Institutes for Biomedical Research, Cambridge, MA, USA. [3]Present address: Max Planck Institute for Physics of the Complex Systems, Dresden, Germany. [4]Present address: Department of Biological Sciences, Birla Institute of Technology and Science-Pilani, Hyderabad Campus, Secunderabad, India. [5]Present address: GRO Biosciences, Cambridge, MA, USA. ✉E-mail: bhattacharyya.sanchari@gmail.com; shakhnovich@chemistry.harvard.edu

complete protein–protein interaction (PPI) map of the *E. coli* 1-carbon metabolism pathway using a biomolecular fluorescence complementation assay. Further, using high-throughput library generation and screening, along with computational simulations using AlphaFold and metadynamics, we answer several key questions about the structural architecture of such transient enzyme assemblies. We discover that most proteins utilize conserved structural interfaces dedicated to interacting with multiple partners, thereby behaving as date hubs as opposed to more permanent multi-enzyme assemblies or party hubs. Finally, using a coarse-grained diffusion-reaction model, we show that such enzyme interactions cause a dramatic enhancement of reaction fluxes that are several orders of magnitude greater as compared to a scenario with no PPI. This study not only represents the first detailed investigation of PPIs forming metabolons in prokaryotic enzyme pathways but also represents the first high-resolution structural insight into the precise molecular architecture of such fleeting miniature protein assemblies.

# Results

## Split YFP system to measure interactions in the 1-carbon metabolism pathway

The 1-carbon metabolism pathway is an essential pathway consisting of 3 inter-dependent biosynthetic pathways that catalyze the transfer of 1-carbon units from methyl donors (5,10-methelene THF, 5-methyl THF, N-formyl THF) to generate purine and pyrimidine nucleotides as well as methionine and pantothenic acid. The methyl donors themselves are produced through the folate biosynthesis pathway that starts with GTP and chorismate and involves condensation of para-amino benzoate (PABA) with 6-hydroxymethyl-dihydropterin diphosphate to form 7,8-dihydropteroate, which is eventually converted to dihydrofolate (DHF) and tetrahydrofolate (THF) (Fig. 1A). The pyrimidine biosynthesis pathway intersects the folate pathway at its second last step catalyzed by thymidylate synthase (ThyA), where uridine monophosphate (dUMP) accepts a methyl group from 5,10-methylene THF to convert to thymidine monophosphate (dTMP). The purine biosynthesis pathway utilizes folate derivatives twice: in the third step where the enzyme PurT catalyzes the conversion of glycineamide ribonucleotide (GAR) to formyl-GAR with the help of formate; in the penultimate step where PurH catalyzes the conversion of AICAR to formyl-AICAR using N-formyl THF as a cofactor.

Our previous work (Bhattacharyya et al, 2016) had shown that enzymes in the folate biosynthesis pathway not only interact among themselves but also with those from functionally related purine and pyrimidine biosynthesis pathways. To systematically map interactions among all proteins in these three related pathways, we used a high-throughput method to measure these interactions. We used a bimolecular fluorescence complementation assay (BiFc) using the split YFP system to measure interaction between pairs of proteins. In this system, one protein is fused to the N-terminus of YFP, while the other protein is fused to the C-terminus. A mature YFP is formed only if proteins A and B interact (Fig. 1B). The two proteins are expressed from two mutually compatible plasmids and using two different inducers (IPTG based, and arabinose based) (Fig. 1C). Stronger interactions give higher fluorescence (Fig. 1E), while weak or non-specific interactions give low fluorescence (Fig. 1D).

## Matrix of 1225 interactions shows a large variation in PPI strengths

Out of the total 41 proteins in the three pathways, genes corresponding to 35 proteins were cloned successfully as both N- and C-YFP fusions, which were used to generate a $35 \times 35$ matrix of 1225 interactions. For each pair, fluorescence was measured in two configurations: NYFP-X/CYFP-Y and NYFP-Y/CYFP-X, where X and Y are the two proteins in question. The entire dataset with interactions in two configurations is shown in Fig. EV1A and Dataset EV2. In many cases, we observed strong interaction only in one configuration (either NYFP-X/CYFP-Y and NYFP-Y/CYFP-X), and not in both. This is presumably because fusion in one configuration may not be optimal for interaction. We therefore generated a symmetric matrix (Fig. 2A and Dataset EV1) by selecting the highest fluorescence value obtained in any of the configurations. For a few pairs, the bacterial cultures co-transformed with both plasmids grew very poorly, hence a significant number of cells could not be analyzed (gray boxes). For the remaining pairs, mean fluorescence intensity (MFI) showed a wide range of variations (Fig. EV2A). As seen from Fig. 2A, the vast majority of the interactions are weak (MFI < 750). Several diagonal elements in the dataset show strong interactions, which correspond to the formation of homo-oligomers. Interestingly, several off-diagonal protein pairs showed highly discernible interaction (MFI > 750). We classified them into weak (blue for $750 < \text{MFI} < 900$), moderate (yellow for $900 < \text{MFI} < 1100$), and strong (red boxes indicate MFI > 1100) interactions.

The BiFc method can yield false positives and the YFP truncated at positions 155/156 used in this study has been shown to have a moderate tendency for self-assembly (Ohashi et al, 2012); hence, we sought to benchmark the method in our case by comparing our results with two existing datasets. First, surface plasmon resonance studies in our previous work (Bhattacharyya et al, 2016) had shown that *E. coli* DHFR binds to PurH and GlyA; however, it shows no detectable binding to adenylate kinase (Adk). The split YFP data in this study show high fluorescence for the FolA-GlyA pair (1002) and for the FolA-PurH pair (917); however, much lower fluorescence for the FolA-Adk pair (641), thereby corroborating the data obtained with purified proteins. Second, Fig. 2A shows several high fluorescence values along the diagonal of the matrix. To find out if this accurately represents the propensity of proteins for homo-oligomer formation, we plotted the fluorescence intensities for known cases of monomeric and oligomeric proteins (dimers and above). As shown in Fig. EV2B, the distributions of MFI for known monomers and homo-oligomers were significantly different ($p$ value = 0.03; the differences become more strongly significant with $p$ value = 0.005 if the case of homo-dimeric PurT with MFI 175 is excluded from the distribution; given that in most cases, absence of interaction or weak interactions are characterized by MFI in the range of 500–650, MFI of 175 seems more likely to be an outlier). Overall, Fig. EV2B provides support that the fluorescence complementation method in our study does not largely lead to false positives.

Since the N- and C-YFP fusion proteins are induced using IPTG and Arabinose, we next attempted to find out how the fluorescence intensity changes as a function of protein expression levels (or inducer concentration). As expected, we found that the increase in inducer concentration causes a sharp increase in YFP fluorescence (Fig. 2B–D), indicating that the observed fluorescence is strongly concentration dependent.

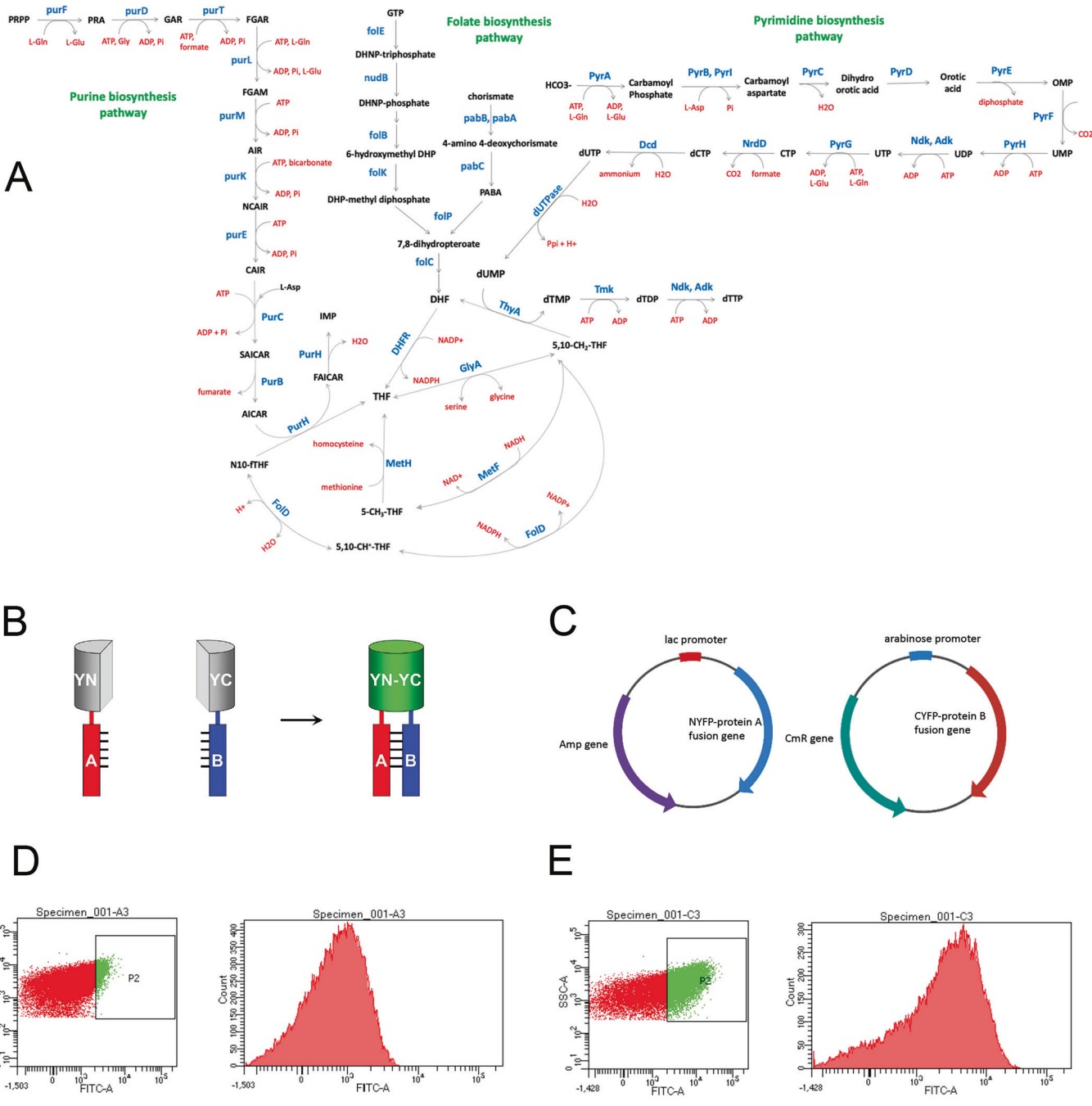

**Figure 1. Overview of the *E. coli* 1-carbon metabolism pathway and YFP complementation method used to detect protein–protein interaction (PPI).**

(A) Purine, pyrimidine, and folate biosynthesis pathways converge onto the 1-carbon metabolic pathway. (B) PPI is detected by bimolecular fluorescence complementation assay (BiFc) using the split YFP system wherein the N-terminus of the YFP molecule is fused to protein A, while the C-terminus is fused to protein B. A mature fluorescent YFP molecule is formed by complementation only if protein A and protein B interact. (C) A mutually compatible two-plasmid-based system used for the expression of YFP-fusion proteins. NYFP-protein A is expressed from an IPTG-inducible lac promoter, while CYFP-protein B is expressed from an arabinose-inducible pBAD promoter. Representative flow-cytometry data for (D) a weakly interacting PPI pair and (E) a strongly interacting pair. For both (D, E), the left panel shows the SSC (side scatter of cells) as a function of fluorescence. SSC is a measure of the internal granularity of the cell, which helps to identify a particular cellular population. P2 highlights the population that shows higher fluorescence than untransformed *E. coli* cells. This population is for visualization purposes only and is not used to calculate any relevant parameter for this study. The histogram on the right panel is derived from all cells (red and green dots together) in the left panel, and the mean fluorescence intensity (MFI) of this histogram is used to assess the extent of interaction.

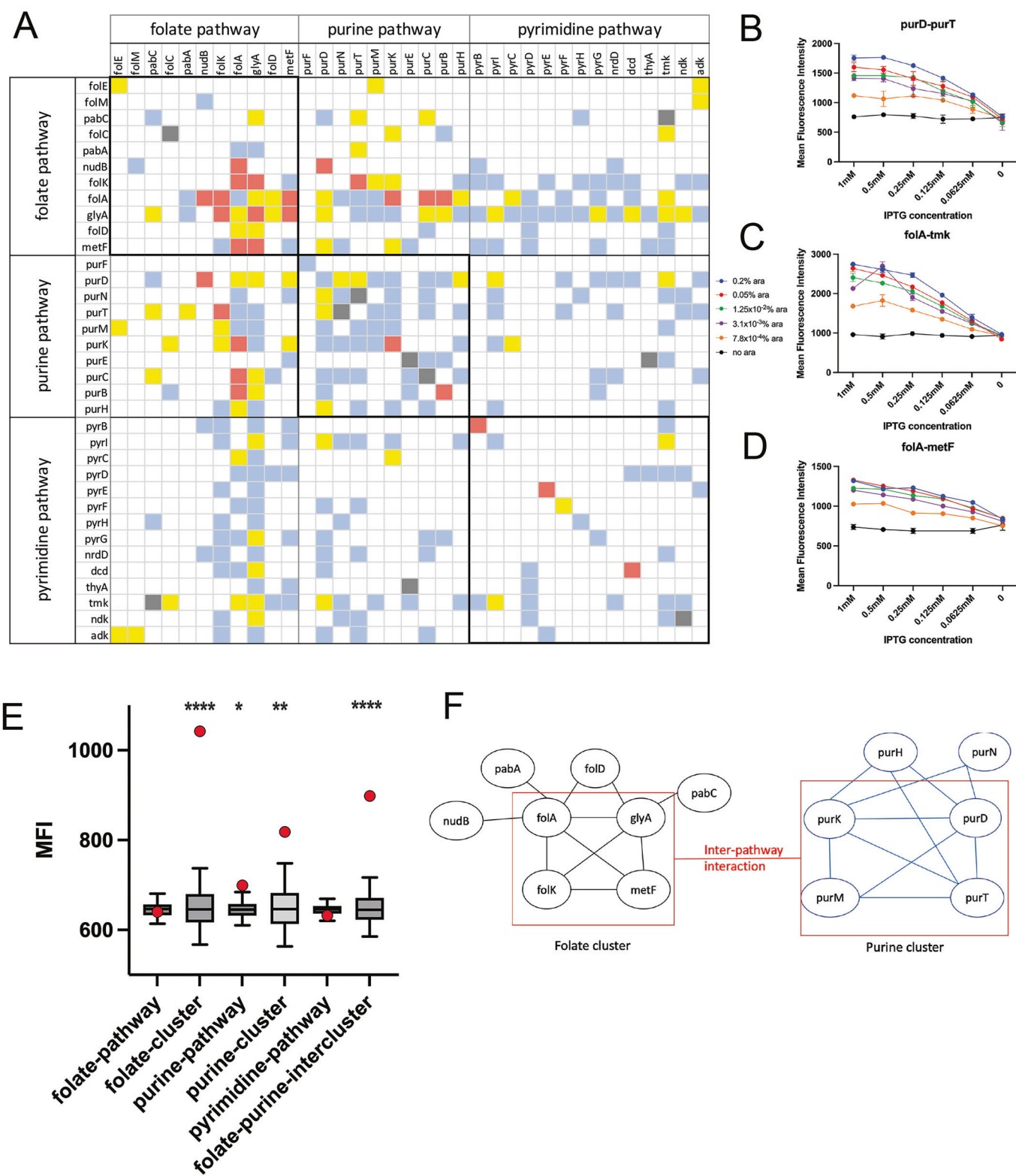

## Folate and purine biosynthesis pathways show multiple strong intra- and inter-pathway interactions

We observed that certain proteins (FolK, FolA, GlyA, and PurD) have a statistically higher propensity to be involved in PPI than others (Fig. EV2D). Next, we sought to find out if folate, purine, and pyrimidine biosynthesis pathways have significant intra-pathway interactions. Our results show that while the entire pathways do not have significantly stronger interactions than random pairs (folate, purine and pyrimidine pathways in Fig. 2E),

**Figure 2. Exploration of the complete PPI network shows a strong intra- and inter-pathway cluster of interactions.**

(A) A total of 35 proteins from the three different pathways were cloned as both N-terminal and C-terminal YFP fusions, and the interaction of every protein with every other protein (including itself, which indicates homodimer probability) was probed using the split-YFP system. The matrix was color-coded based on the mean YFP fluorescence of the pair. The interaction was evaluated in both formats (NYFP-A/CYFP-B and NYFP-B/CYFP-A), and the maximum fluorescence data for each pair were used to generate this matrix. Red boxes indicate MFI > 1100, yellow for 900 < MFI < 1100, blue for 750 < MFI < 900, while white boxes indicate MFI < 750. Titration of inducer levels (arabinose and IPTG) to test dependence of fluorescence values on intracellular protein expression for (B) purD-purT, (C) folA-tmk, and (D) folA-metF pairs. As expected, higher inducer concentration leads to more complex formation and stronger YFP fluorescence. Data is presented as the mean of $n = 2$ independent biological replicates, and the error bars indicate standard deviation (s.d.). (E) While there is no significant clustering for all 11 proteins in the folate pathway ($n = 55$ pairs of interactions), a set of 5 proteins ($n = 10$ PPI pairs for FolK, FolA, GlyA, FolD, and MetF) shows strong clustering ($p$ value = 0.0001). Similarly, a set of 5 proteins in the purine biosynthesis pathway (PurK, PurD, PurM, PurD, PurN) shows cluster formation ($n = 9$ PPI pairs) ($p$ value = 0.005 vs $p$ value = 0.02 for all proteins in the purine biosynthesis pathway). While no statistically significant clustering is observed among pyrimidine biosynthesis pathway proteins, there is strong evidence of inter-pathway interaction among selected folate and purine biosynthesis enzymes ($p$ value = 0.0001). For each cluster, the red data point represents the median MFI value of the cluster, while the box represents a null distribution of the median values of an equal number of interaction pairs (as the cluster size) randomly picked 10,000 times from the matrix. The box represents the 25–75 percentile of the data, and the line in between represents the median of the distribution. The whiskers represent the 5–95 percentile interval. Statistical significance and $p$ value are calculated using a one-sided non-parametric permutation test as described in "Methods," *** indicates $p$ value < 0.001, ** indicates $p$ value < 0.01. No data normality is assumed. (F) A pictorial graph representation of the strongest interactions. Corners represent protein identity while edges represent strong interaction. Source data are available online for this figure.

smaller clusters within the pathways do exhibit strong interactions that are statistically significant (folate cluster and purine cluster in Fig. 2E). For example, five proteins of the folate pathway, namely, FolK, FolA, GlyA, FolD, and MetF, interact strongly with each other, presumably forming an intra-pathway cluster (folate-cluster in Fig. 2E, which refers to ten PPI pairs among the five proteins listed above). Interestingly, these are also sequential proteins in the folate pathway. Similarly, several proteins of the purine biosynthesis pathway (namely, PurD, PurN, PurT, PurM, and PurK) also form an intra-pathway cluster (purine cluster in Fig. 2E), though the interactions are overall weaker than those of the folate pathway. No such cluster was formed among enzymes from the pyrimidine biosynthesis pathway. The only exception in the pyrimidine pathway is thymidylate kinase (Tmk), which interacts with several proteins within its own pathway, along with proteins from purine and folate pathways (Fig. EV2D). Interestingly, it was shown that Tmk overexpression is toxic, probably due to mis-interactions upon overexpression with other proteins of the folate pathway (Bhattacharyya et al, 2021; Bhattacharyya et al, 2016). Next, we checked for significant inter-pathway interactions among the three pathways. While overall there are no statistically significant inter-pathway interactions when pathways are taken as a whole (Fig. EV2C), certain clusters of proteins in the folate and purine biosynthesis pathway do show statistically significant inter-pathway interactions (16 PPI pairs referred to as folate–purine inter-cluster in Fig. 2E). Based on the strong interactions observed, we constructed a graphical representation of possible clusters within the pathways and between folate and purine pathways (Fig. 2F). According to this, FolA, FolK, GlyA, and MetF form a core cluster in the folate pathway, where each protein interacts with every other protein. FolD, PabA, PabC, and NudB proteins interact with one or two of the core proteins, but do not interact among themselves; hence, they form peripheral members of the cluster. A similar analysis shows a core cluster in the purine biosynthesis pathway, comprising PurK, PurD, PurM, and PurT, while PurH and PurN are peripheral members. We also calculated the statistical significance of the clusters identified in Fig. 2E based on the original matrix of interactions in both configurations (Fig. EV1A). As seen in Fig. EV1B, the clusters still remained highly statistically significant, though the mean MFI of the cluster is lower. This is expected since the interaction may not be optimal in one configuration.

One possibility is that the pronounced number of interactions for certain proteins (e.g., FolA, GlyA, FolK, PurD) observed in this work is due to other properties that are unrelated to functional interactions. For example, it might be that these subsets of proteins accumulate in inclusion bodies at the poles or simply have higher abundance than the other proteins, as YFP fluorescence is indeed strongly dependent on protein expression levels. To that end, we carried out a western blot analysis to quantify expression levels of the 35 N- and C-YFP fusion proteins (Fig. EV3). The overall expression levels of the CYFP-fusion proteins were less than those of the NYFP fusions. Though there were some variations in the expression levels within one set, they were not substantial enough to explain the variations in fluorescence intensities. For example, NYFP-FolA that shows strong interactions with multiple proteins (Fig. 2A) had comparable expression to NYFP-PyrE and NYFP-Dcd; however, PyrE and Dcd do not show any significant PPI. On the other hand, NYFP-Ndk, which has the highest abundance in the dataset, shows poor interaction with most partner proteins. MetF, which has low abundance in both NYFP and CYFP fusions, in fact shows strong interactions with several proteins in the pathway. Overall, this suggests that the high interaction propensity of certain proteins in the pathway is not due to their higher abundance.

Previous studies had clearly shown that over-expressed DHFR protein (FolA) does not accumulate in inclusion bodies to any substantial degree (Bershtein et al, 2013). To confirm this in the present study, we looked at the distribution of fluorescence intensity in live cell imaging experiments. Appendix Fig. S1 shows that in two representative strong interaction pairs (FolA-GlyA and PurD-PurT), the fluorescence was uniformly distributed across the cells with no accumulation at the poles, indicating that the observed YFP fluorescence is not biased due to protein localization at the inclusion bodies.

## Purine biosynthesis interactome is made up of structurally similar proteins

We next asked, are there physico-chemical characteristics of the enzymes that dictate their interactions? Theoretical studies have shown that structurally similar proteins tend to interact (Lukatsky et al, 2007). To that end, we computed the structural similarity index of all proteins against all others using a structural similarity

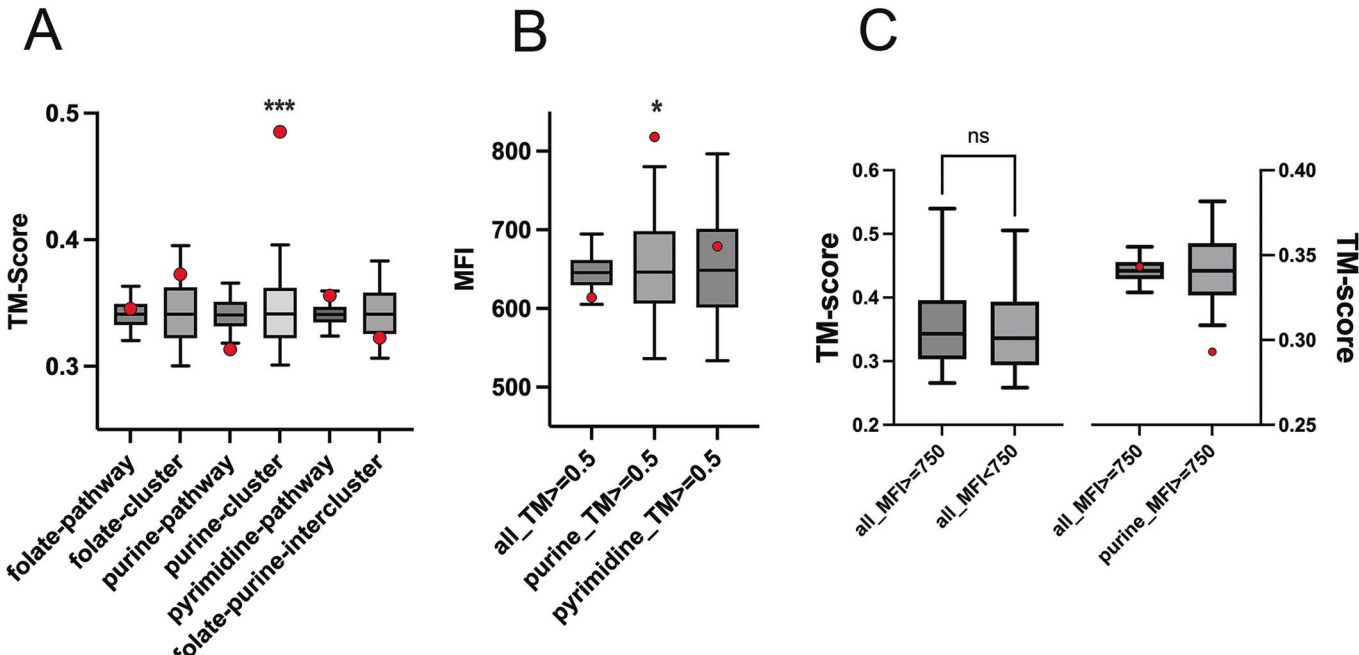

**Figure 3. Subgroup of purine biosynthesis pathway proteins that interact strongly also have strong structural similarity.**

(A) shows median TM scores (red data point) for all groups presented in Fig. 2E (all proteins in folate, purine, and pyrimidine pathways, as well as the clusters) compared to a null distribution of TM values from the matrix (Dataset EV3). Only the purine cluster (PurD, PurN, PurT, PurM, and PurK, total nine pairs) has a highly statistically significant TM score (p value = 0.0004, *** indicates p value < 0.001), while none of the other pathway proteins show any significant structural similarity. The null distribution is generated from the median values of an equal number of interaction pairs (as the cluster size) randomly picked 10,000 times from the matrix (represented by the box plot, where the box represents 25–75 percentile of the data and line in between represents the median of the distribution. The whiskers represent the 5–95 percentile interval). Statistical significance and p value are calculated using a one-sided non-parametric permutation test as described in Methods. No data normality is assumed. (B) Median fluorescence intensities for all PPI pairs that show strong structural similarity (TM score >0.5). The data shows that while a high TM score by itself does not guarantee interaction, those in the purine biosynthesis pathway show mildly significant interaction propensity (p value = 0.022). The null distribution and p values are calculated in the same way as (A). In the box-plot, the box represents 25–75 percentile of the data and line in between represents the median of the distribution. The whiskers represent the 5–95 percentile interval. (C) A similar analysis showing median TM scores for all protein pairs that show strong interaction in the split YFP assay. The results indicate that structural similarity is not a general mechanism that leads to stronger interaction. On the left panel, the distributions of all_MFI > = 750 (n = 142) and all_MFI = < 750 (n = 453) are compared using an unpaired Welch t test, assuming normal distribution of data. On the right panel, similar to (A, B), the red data point represents the cluster median, and the box plot represents the null distribution. p values are calculated using a one-sided non-parametric permutation test as described in "Methods." For all box plots, the box represents 25–75 percentile of the data and line in between represents the median of the distribution. Source data are available online for this figure.

measure, TM score (Zhang and Skolnick, 2005) (Appendix Fig. S2 and Dataset EV3). We first asked if the protein clusters identified from fluorescence data are structurally similar or not. We found that neither the entire folate pathway nor the five proteins in the folate cluster (FolK, FolA, GlyA, FolD, and MetF, total ten pairs) have a significantly higher median TM score compared to a null distribution (Fig. 3A). On the contrary, the set of five proteins from the purine cluster (PurD, PurN, PurT, PurM, and PurK, total nine pairs) has a highly statistically significant greater TM score distribution compared to the null distribution (p value < 0.001) (Fig. 3A), though the ten proteins in the purine pathway as a whole does not share any significant structural similarity. The folate–purine inter-pathway cluster (16 interactions) also has no significant structural similarity. It is worth mentioning at this point that previous studies demonstrated that certain enzymes of the purine biosynthesis pathway evolved from a single common ancestor by gene duplication and divergence (Kappock et al, 2000; Zhang et al, 2008). Thus, structural similarity within this subset of purine biosynthesis pathway proteins is likely to be a factor in selecting them as an interacting cluster in the metabolon

rather than a mere consequence of their origin as divergently evolved from a common ancestor.

To understand this further, we analyzed the fluorescence intensity of all those PPI pairs that have significant structural similarity (TM score >0.5), both in the extended pathway as well as within individual pathways. Additionally, we also analyzed TM score distributions of all PPI pairs that have MFI values >750 (Fig. 3B,C). The results show that while a higher TM score overall does not guarantee interaction, it does dictate interaction propensity in the purine biosynthesis pathway.

## Saturation mutagenesis reveals dedicated PPI interfaces on DHFR that are conserved between its PPI partners

Since enzymes in a pathway interact via weak and transient interactions, no structural information is available yet for such PPI pairs by traditional methods. We reasoned that using our split YFP-based methodology to screen for binding, we could highlight mutational hotspots in the PPI pairs that render these complexes stronger or weaker. To that end, we focused on two PPI pairs,

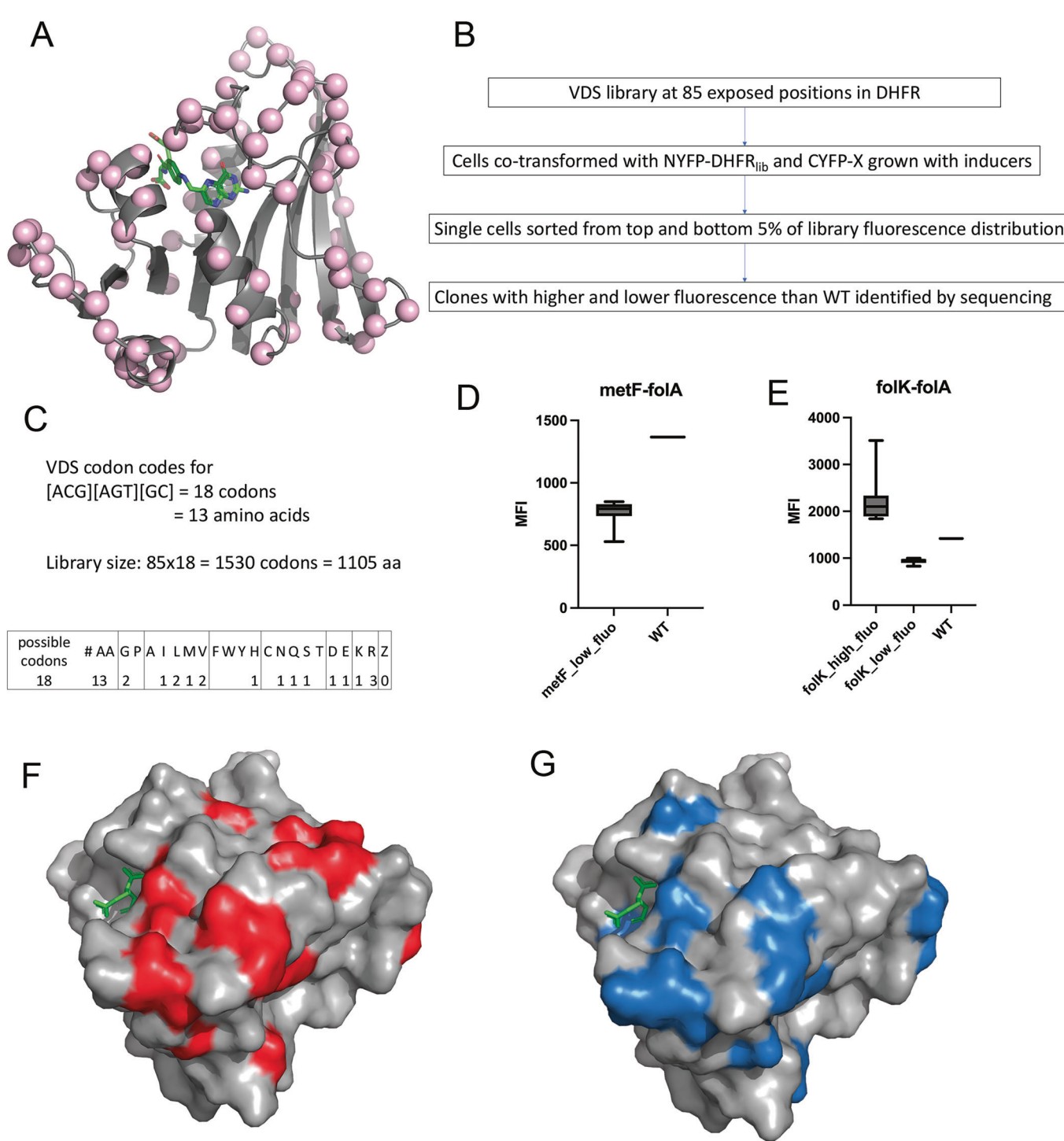

A

B

VDS library at 85 exposed positions in DHFR

Cells co-transformed with NYFP-DHFR$_{lib}$ and CYFP-X grown with inducers

Single cells sorted from top and bottom 5% of library fluorescence distribution

Clones with higher and lower fluorescence than WT identified by sequencing

C

VDS codon codes for
[ACG][AGT][GC] = 18 codons
                = 13 amino acids

Library size: 85x18 = 1530 codons = 1105 aa

| possible codons | # AA | G | P | A | I | L | M | V | F | W | Y | H | C | N | Q | S | T | D | E | K | R | Z |
|---|---|---|---|---|---|---|---|---|---|---|---|---|---|---|---|---|---|---|---|---|---|---|
| 18 | 13 | 2 | | 1 | 2 | 1 | 2 | | | | 1 | | 1 | 1 | 1 | | 1 | 1 | 1 | 3 | 0 |

D   **metF-folA**

E   **folK-folA**

F

G

FolA-MetF and FolA-FolK, which form one of the strong complexes in our dataset. We selected all surface-exposed residues of FolA (accessibility >80%, shown by pink spheres in Fig. 4A), except its active site and contacting residues (within 5 Å of the active site), and generated a near-saturation mutagenesis library using VDS codon. VDS codes for all different amino acid types (hydrophobic, polar, charged), however does not incorporate large hydrophobics (F, W, or Y) or proline, and does not incorporate any stop codon (Fig. 4C). This is to ensure that the change in

fluorescence (if any) will not arise due to protein destabilization or due to a truncated protein. When *E. coli* cells were co-transformed with NYFP-FolAlib and CYFP-MetF/CYFP-FolK (Fig. 3B), the overall histogram of fluorescence intensity of the library variants did not appear to be different from wild type (WT) (Appendix Fig. S3). On the other hand, single cells sorted from the top and bottom 5% of the distribution were found to have a large variation in fluorescence intensities. We selected 33 clones for the DHFR-MetF interaction and 28 clones for the DHFR-FolK interaction that had

**Figure 4. Elucidation of PPI binding interface using saturation mutagenesis-based library screening.**

At each of the 85 solvent-exposed locations (% accessibility >20%) in DHFR protein (Cα residue shown by a light pink sphere in (**A**)), a VDS codon was introduced (one position at a time) by PCR. (**C**) VDS codes for 18 codons (13 different amino acids, no stop codons), which results in a total library size of 1105 amino acids. (**B**) shows a flow-chart of the method used to map the binding interface. The NYFPlib plasmid was transformed with either CYFP-MetF or CYFP-folK expressing plasmids. Single cells were sorted out from the top and bottom 5% of the fluorescence histogram of the library variants, and clones with MFI significantly lower or higher than WT DHFR were sequenced to reveal their identities. (**D, E**) represent boxplots of MFI for the sequenced clones. The box represents 25–75 percentile of the data and line in between represents the median. The whiskers extend from the minimum to the maximum values of the dataset. $n = 33$ for MetF_low_fluo, $n = 17$ for FolK_high_fluo, and $n = 22$ for FolK_low_fluo. The mutations were mapped back on the DHFR structure, and surprisingly, the two interaction surfaces show very significant non-random overlap ($p$ value $= 2e{-}7$ obtained using a Student's $t$ test, see "Methods" and text) that is away from the folate binding site (DHF shown in green sticks). (**F, G**) are DHFR structures where the MetF and FolK binding residues identified from the library screening are shown in red and blue colors, respectively. Source data are available online for this figure.

considerably higher or lower fluorescence intensities than the WT pair (Fig. 4D,E), and sequenced them to reveal the identities of the mutants. Surprisingly, we found that a vast majority of the residues overlap between FolA surfaces interacting with FolK and MetF (26, 29, 114, 124, 131, 132, 140–145, 149, 157, 159) and are highlighted in Fig. 4F,G. When these mutational hotspots were mapped back onto the structure of DHFR, we found that the binding site is mostly located in the C-terminal beta hairpin, as well as in the helix close to the beta hairpin. The binding site does not overlap with the active site (folate or NADPH binding site).

To test whether there is a non-random, statistically significant overlap between the FolA interaction surfaces with Folk and MetF (Fig. 4F,G), we applied statistical test described in "Methods" and found that DHFR uses the same surfaces to interact with MetF and FolK: the overlap between two interaction surfaces is highly significant with a $p$ value of $10^{-7}$ (see "Methods").

A possible alternative explanation for the drop in fluorescence in mutants is that mutations could lead to loss of DHFR protein abundance due to unfolding/misfolding/aggregation caused by DHFR destabilization. To address this possibility, we performed a western blot analysis of the mutants with anti-DHFR antibodies. None of the mutations showed any substantial change in intensity/abundance (Appendix Fig. S4), indicating that the mutations genuinely perturb binding without affecting abundance.

## Raman spectroscopy of purified proteins confirms binding site residues

Next, we sought to validate the interaction between DHFR and FolK (encoded by the gene folK) and its affinity-changing mutants revealed in high-throughput mutagenesis. To that end, we performed Raman spectroscopy and specifically looked at the spectral features of the amide I range (1600–1700 cm$^{-1}$) for DHFR and FolK individually and then upon mixing. Amide I spectra originate from the C = O stretch in the peptide backbone and are extremely informative about protein conformations. Analysis of amide I spectra helps in the mechanistic understanding of secondary structures and their organization in a protein. Deconvolution of amide I spectra with standard peaks for discrete secondary structural elements, viz. helices, sheets, and loops, helps in the quantification of secondary structure content in a protein (Gallagher, 2009). We performed deconvolution of the amide I spectra of the DHFR and FolK using Lorentzian peak fitting with pre-assigned peaks to better understand the change in the spectral features upon mixing both protein solutions to detect non-additivity, which would report on the interaction between two

proteins. The quality of fitting was checked with reduced chi-squared values, which were typically around 1.2.

Secondary structure contents obtained from deconvoluted DHFR (Fig. EV4A) and FolK spectra (Fig. EV4B) were compared with secondary structure contents of the crystal structures of DHFR (PDB: 1DRA) and FolK (PDB: 4M5I) to check the spectral quality and physical condition of the samples. Secondary structure contents for both DHFR (Fig. EV4A) and FolK (Fig. EV4B) were found to be comparable to the secondary structure contents of the reference structures.

Upon mixing the DHFR and FolK protein solutions, any change in the amide I spectral feature is a potential indication of spectra originating from the conformation of the bound protein–protein complex, which in turn indicates PPI. In the absence of interaction, spectra appear as a mathematical sum of two individual protein spectra. In our study, we therefore investigated spectral features under conditions of DHFR FolK mixing. Control experiments were performed with ADK mixed individually with DHFR and FolK. We performed a similar quality check experiment with ADK as was performed with DHFR and FolK using the crystal structure of ADK as the reference and observed comparable secondary structure contents.

Upon mixing DHFR and FolK, the resultant amide I spectra showed new features around 1663 cm$^{-1}$ not observed in the individual spectrum of DHFR and FolK before mixing (Fig. 5A). Further, the resultant spectrum was not the mathematical sum of individual spectra obtained with DHFR and FolK before interaction (Fig. EV5A). Interestingly, no new feature in the amide I spectra was observed for control acquisitions where ADK was mixed individually with DHFR (Fig. EV5B) and FolK (Fig. EV5C). The spectra obtained in these control acquisitions were equal to the mathematical sum of DHFR and ADK spectra before mixing in the DHFR-ADK mix and similarly FolK and ADK spectra in the FolK-ADK mix. The presence of a new spectral feature with a peak around 1663 cm$^{-1}$ in the amide I spectral profile of the DHFR-FolK mix (Fig. 5A) is a potential indication of an interaction between FolK and DHFR and resultant amide I spectra potentially reflecting the secondary structural organization of the FolK-DHFR complex in solution.

Next, we carried out Raman spectroscopy using a range of DHFR concentrations (from 7.5 to 50 μM) and keeping FolK concentrations fixed at 15 μM. We observed area under the characteristic peak at 1663 cm$^{-1}$ increases monotonically (Fig. 5B). The changes in the interaction-specific peak area at ~1663 cm$^{-1}$ were plotted against the increasing DHFR concentrations (Fig. 5C). As the peak chosen was found to be specific for the interaction and

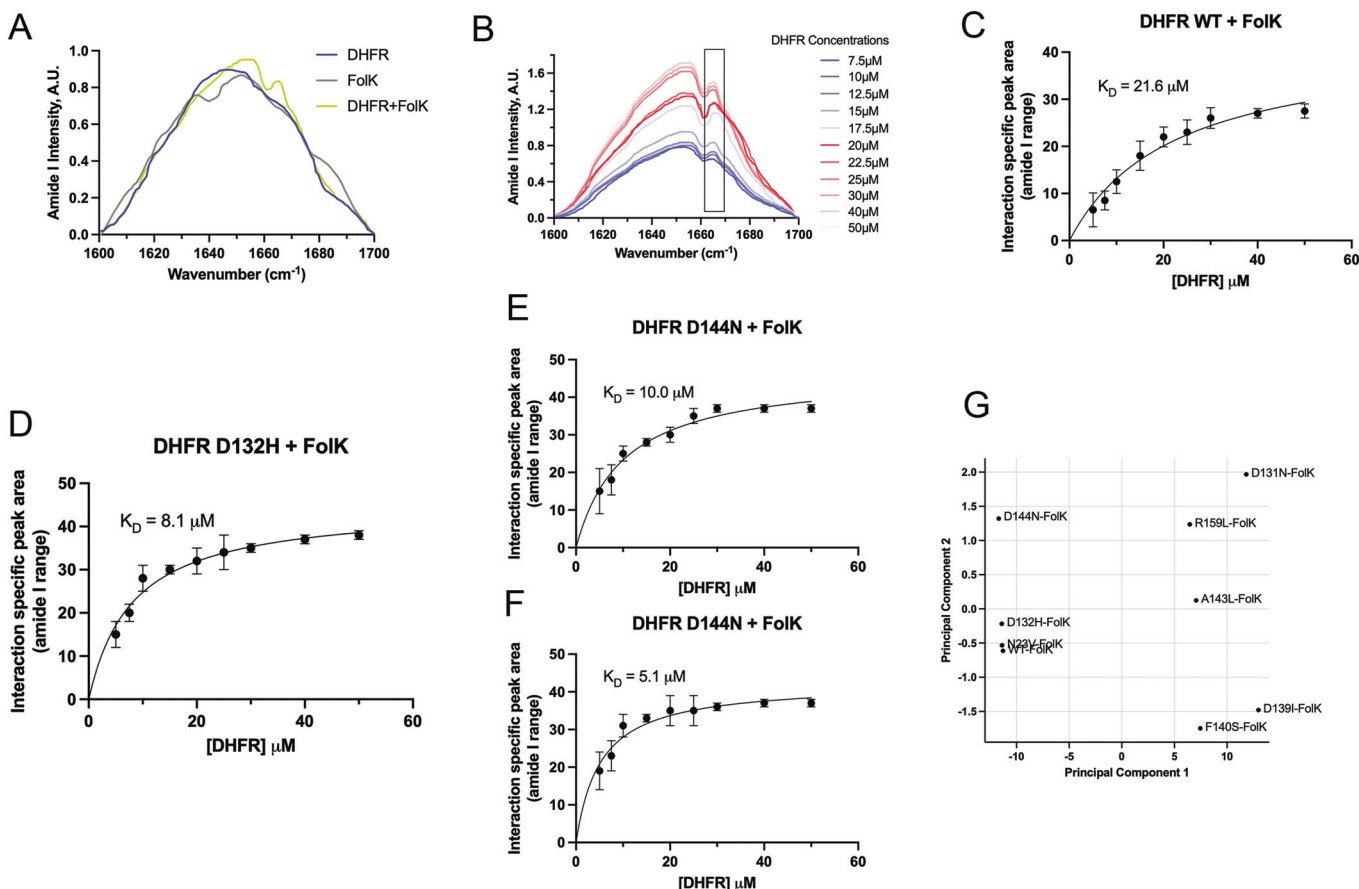

**Figure 5. Raman spectroscopic analysis shows FolK interacts with WT and select mutants of DHFR.**

(A) Amide I Raman spectroscopic profiles of WT-DHFR (blue trace), FolK (gray trace), and DHFR-FolK complex (yellow trace) are shown. Intensities (arbitrary units, Arb. Units) for WT DHFR and FolK in the amide I range (1600–1700 cm⁻¹) were recorded individually and then for the WT DHFR-FolK complex. The spectrum for the WT DHFR-FolK complex is not a mathematical sum of individual spectra recorded for WT DHFR and FolK and shows new spectral features and a characteristic peak at ~1665 cm⁻¹ indicating a signal originating from the WT DHFR-FolK-bound complex. (B) Amide I spectral profiles of WT DHFR-FolK complex along an increasing concentration of DHFR (7.5–50 μM). The marked section shows the peak at ~1665 cm⁻¹ characteristic of WT DHFR-FolK interaction. (C) Changes in the peak area at ~1665 cm⁻¹, characteristic of WT DHFR-FolK interaction, were plotted against increasing concentrations (7.5–50 μM) of DHFR. The peak areas were obtained upon deconvolution of the spectral read-outs ($n = 3$), and the plotted data were subjected to hyperbolic fitting to derive the WT DHFR-FolK dissociation constant, $K_d$, for WT DHFR-FolK interaction, which was found to be 22 μM. (D) Changes in the peak area at ~1665 cm⁻¹, characteristic of D132H mutant of DHFR-FolK interaction, were plotted against increasing concentrations (7.5–50 μM) of D132H DHFR. $K_d$ for D132H DHFR-FolK interaction was found to be 8.1 μM. (E) Changes in the peak area at ~1665 cm⁻¹, characteristic of D144N mutant of DHFR-FolK interaction, were plotted against increasing concentrations (7.5–50 μM) of D144N DHFR. $K_d$ for D144N DHFR-FolK interaction was found to be 10 μM. (F) Changes in the peak area at ~1665 cm⁻¹, characteristic of N23V mutant of DHFR-FolK interaction, were plotted against increasing concentrations (7.5–50 μM) of N23V DHFR. $K_d$ for N23V DHFR-FolK interaction was found to be 5.1 μM. (G) Principal component analysis (PCA) based on the entire Raman spectroscopic profiles (800–2000 cm⁻¹) for WT and 8 mutants of DHFR. The positions of the mutants in the plot indicate their relatedness based on their Raman profiles. Mutants that are closely positioned on the plot share similarities in their Raman spectra. PCA shows FolK-interacting mutants, viz. D132H, N23V, and D144N along with the WT are clustered closely. On the other hand, non-FolK mutants of DHFR are separately spaced. $n = 3$ biologically independent samples were used for the experiments. All the data are presented as mean values +/− SEM. Source data are available online for this figure.

no non-specific/unbounded fraction-specific signal was included in the plot, the hyperbolic model was used to fit the data. $K_d$ was obtained upon hyperbolic fitting of the data and was found to be 22 μM for WT DHFR-FolK interaction (Fig. 5C). Quality of the model deployed for fitting the data was assessed by $R^2$ value, and it was found to be 0.9. Next, we carried out similar interaction analyses with FolK for select mutants of DHFR that are predicted to weaken or strengthen binding based on FACS data (D144N, N23V, and D132H increase fluorescence, while D131N, E139I, F140S, A143L, and R159L showed loss in fluorescence). Interestingly, the D132H mutant of DHFR did show a stronger interaction with FolK

with a $K_D$ of 8.1 μM (Fig. 5D). Similarly, compared to WT DHFR, D144N and N23V mutants of DHFR were also found to have a stronger interaction with FolK with $K_D$ values of 10 and 5.1 μM, respectively (Fig. 5E,F). For mutants that show a loss in fluorescence in the FACS analysis (D131N, E139I, F140S, A143L, R159L), quite strikingly, the characteristic peak at 1663 cm⁻¹ was absent upon interaction, and hence an exact $K_D$ could not be derived. While this directly hints at loss in binding, we attempted to compare more global signatures of the Raman spectra among mutants; hence, we resorted to principal component analysis (PCA). We performed PCA on the full Raman spectral profiles

recorded across the 800–1900 cm$^{-1}$ range. This comprehensive spectral window includes not only the amide I region, which is commonly associated with secondary structural elements, but also broader backbone and side-chain vibrational features that can capture higher-order structural organization. PCA enables the reduction of this high-dimensional dataset into orthogonal principal components that account for the greatest variance, thereby facilitating an unbiased, global comparison of spectral signatures associated with each DHFR-FolK mixture.

In the resulting PCA projection (Fig. 5G), samples corresponding to the WT DHFR-FolK complex and those involving high-affinity DHFR variants (D132H, D144N, N23V) form a coherent and tightly clustered group in PC1–PC2 space, indicating reproducible and structurally similar conformational changes associated with complex formation. In contrast, the profiles corresponding to DHFR variants previously shown to lose binding capacity (D131N, E139I, F140S, A143L, R159L) are dispersed away from this cluster, highlighting distinct spectral trajectories and supporting the absence of complex-specific conformational rearrangement. Importantly, this analysis does not rely on any single spectral feature—such as the emergence of the 1663 cm$^{-1}$ DHFR-FolK interaction-specific peak—but captures cumulative differences across the entire spectrum, offering a robust validation of DHFR-FolK complex formation. PCA results not only confirm the presence of an interaction-specific signature in the DHFR-FolK complex but also establish that the resultant spectral profiles are not a mathematical sum of the components. PCA results provide an independent and comprehensive demonstration that the observed spectral patterns in interacting DHFR-FolK mixtures represent non-additive, interaction-specific conformational signatures, beyond what can be explained by simple peak-based analysis.

## High-throughput computational approach reveals that enzymes use similar interfaces to interact with multiple proteins

Mutagenesis-based library screening experiments so far reveal that the FolA protein interacts with two distinct partners, FolK and MetF with remarkably similar interaction interfaces. However, due to experimental limitations, the interaction interface study could only be performed for a limited number of interaction partners. To study whether multiple folate pathway proteins use a similar interface to interact with each of their PPI partners, we employed a high-throughput computational approach using AlphaFold3. We first use AlphaFold3 to predict dimeric structures, for all pairwise interactions between different members of the folate pathway (Fig. 6A), like the experimentally generated matrix in Fig. 2A. Subsequently, we performed metadynamics simulations (see detailed description in "Methods") to further refine these dimeric structures and their corresponding interaction interfaces (Fig. 6A). To elucidate the most stable PPIs, we identify pairwise inter-chain residue-level contacts for all structures that correspond to free energy minima identified in the metadynamics simulations (Fig. 6A). We also compute binding free energies ($\Delta G_{binding}$) from the free energy profiles obtained using our metadynamics simulations. In Fig. 6B, we compare whether the protein pairs computationally predicted to be stable binders ($\Delta G_{binding} < 3$ kcal/mol) also show stable binding in experiments. For the sake of comparison, dimers predicted to be stable binders in simulation were considered as "true predictions" if the MFI for the

corresponding protein pair in the fluorescence experiments was greater than 670. We note that more than 65% of the simulation predictions are "true predictions" for a range of $\Delta G_{binding}$ and experimental fluorescence thresholds (also see Fig. EV6). The robustness of simulation predictions is consistent across the three pathways, and for a range of experimental and simulation thresholds (Figs. 6B and EV6). These results suggest that our metadynamics simulation protocol can be reliably employed to screen PPIs and identify stable binders. It must be noted that such an approach for comparing experimental fluorescence values was chosen over a direct correlation analysis between the binding free energies from simulation and fluorescence because the two techniques could have very different dynamic ranges. In other words, an increase in the $\Delta G_{binding}$ computed from the metadynamics simulations might not directly correlate to a commensurate increase in fluorescence values in experiments, making the exercise of computation of a direct correlation somewhat misleading. The simulation data is for simple two-component binding, unlike the experimental values which could be perturbed by the presence of other molecules in solution. Therefore, weak, stable binders in metadynamics simulations might not directly show up in fluorescence signals in experiments at biologically relevant concentrations in a more complex cellular milieu. The free-energy minimum structures from our simulations are further used for analysis of interaction interfaces.

In Fig. 6C, we then compare the extent of overlap in predicted interaction interfaces of the FolA (from metadynamics simulations) for the FolA-FolK pair and compare it to the experimentally determined interface. Figure 6C shows that the simulation predictions agree with the experimentally predicted interaction interfaces, suggesting that simulations can be used to screen interaction interfaces in a high-throughput fashion. The overlap between the experimental and simulation predicted surfaces is statistically significant with a $p$ value of $2.3 \times 10^{-5}$, under the null hypothesis that computation and experiment pick two unrelated surfaces on FolA (see "Methods").

Interestingly, structural alignment of the AlphaFold3 predicted complexes of FolA with its strongest partner proteins NudB, FolK, GlyA and FolD clearly shows that different proteins bind around a similar region of FolA (Fig. EV7A). To gain a more quantitative insight into this observation for all folate pathway proteins, we calculate the frequency of involvement of every amino acid residue in heteromeric PPIs across different partners within the pathway. In Fig. 7A, we represent the AlphaFold3 predicted structures of folate pathway proteins with residues that are frequently involved in PPI across different partners represented as large (yellow to green colored) spheres while those that are less involved in PPIs are represented as smaller, darker colored spheres. As seen from Fig. 7A, the residues most frequently involved in heterodimeric protein–protein contacts across all their PPI partners are clustered in a small region of the protein, as evident from the spatial segregation of bright and dark colored spheres on the folate pathway proteins in Fig. 7A.

To check if there is a non-random overlap of interaction surfaces across different interaction partners, we employ the statistical analysis (see "Methods" section for details) that provides $p$ value of observed overlaps between interaction surfaces with different interaction partners against the null hypothesis that a protein utilizes unrelated surfaces to bind to different interaction partners in the folate metabolon (also see Appendix Fig. S5). In

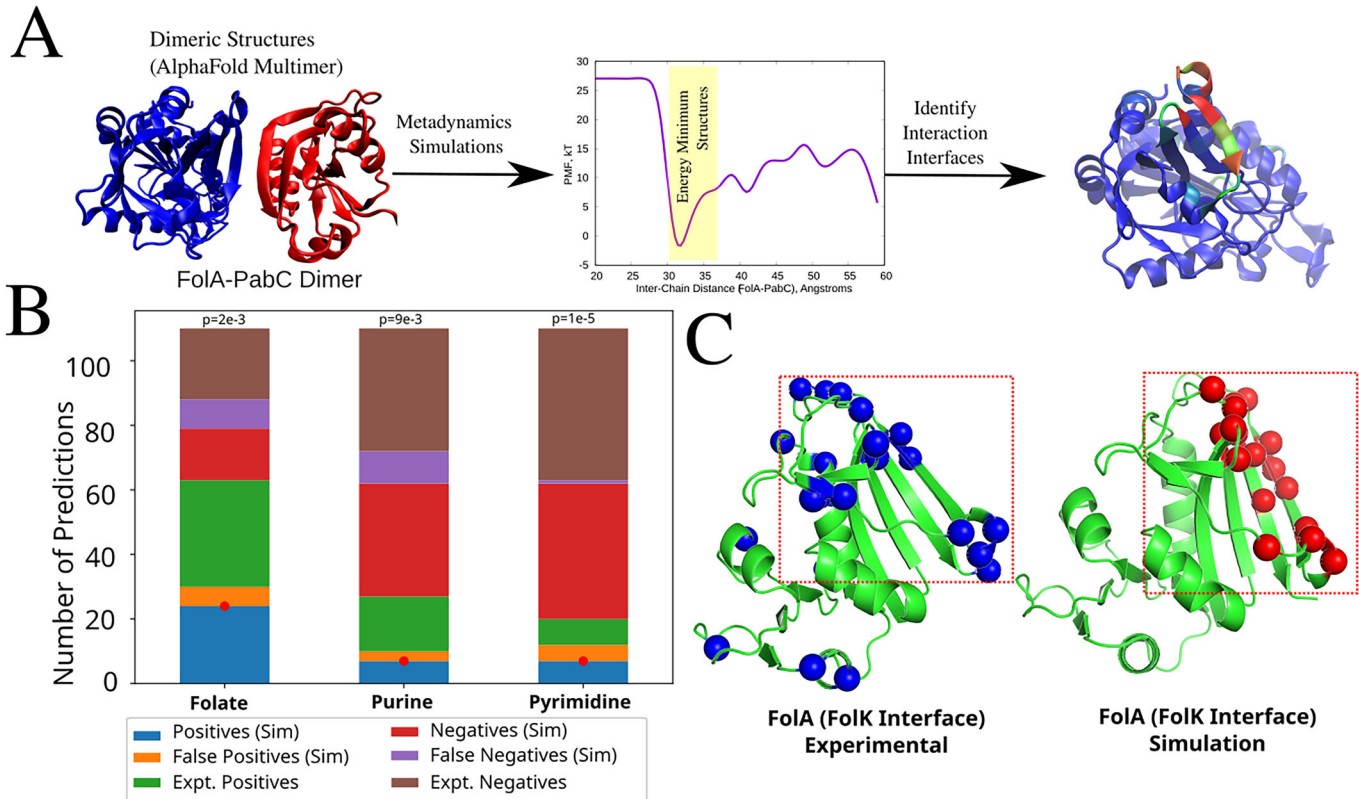

**Figure 6. Metadynamics simulations used to compute interaction interfaces and unbinding free energies for folate pathway dimers.**

(A) AlphaFold 3.0 is used to make initial dimeric structure predictions for all possible folate pathway enzyme pairs. To further refine these structures and compute their stabilities, metadynamics simulations were performed, with AlphaFold3 predicted structures as starting points and with inter-chain distance as the collective variable. The structures corresponding to the free energy minima in the PMFs were then used to study the interaction interfaces for folate pathway proteins. (B) Dimers with $\Delta G_{binding} < -3$ kcal/mol in metadynamics were considered stable binders. Those dimers predicted to be stable binders in simulation were considered as "true predictions" if the MFI for the corresponding protein pair in the fluorescence experiments was greater than 670. $p$ values above each bar indicate statistical significance of agreement between theory and experiment against the null hypothesis that simulations predict binding surfaces between pairs of proteins at random. A more complete analysis of agreement between computational predictions and experiment for a range of thresholds is in Fig. EV6. (C) Simulations and experiments predict a similar interaction surface for FolA with FolK. Residues of FolA involved in interactions with FolK based on experimental data (marked as red beads in the left structure) and metadynamics simulations (marked as cyan beads in the right structure). The simulation predicted surface and the experimentally detected interaction surfaces show significant overlap ($p$ value of $2.7 \times 10^{-5}$ for the null hypothesis that experimental and predicted interaction surfaces have random overlap, see "Methods" for details). Source data are available online for this figure.

Fig. 7B, we present the results of this analysis in the form of the ratio of mean inter-surface contacts for the observed surfaces (from metadynamics simulations) vs that of control pairs of randomly drawn surfaces on the protein. A high ratio in Fig. 7B suggests that the observed surfaces share a high degree of overlap compared to what would be expected for randomly chosen surfaces on the protein. As evident from the $p$ values shown in Fig. 7B, we observe a statistically significant overlap in interaction surfaces across different interactions for all proteins of the folate pathway, except FolE and FolD. The corresponding Z-scores are presented in Appendix Fig. S5. Interestingly, despite being involved in interactions across multiple different partners, similar interfaces on the protein get shared across several different folate-pathway interaction partners. The metadynamics simulations, therefore, suggest that these proteins could exhibit a great deal of promiscuity in PPIs, with the same interface being involved in interactions across multiple different partners. A remarkable exception seems to be FolD, which, despite being involved in strong interactions with

FolA and GlyA, does not seem to involve a common interaction interface. Structural alignment (Fig. EV7B) shows that FolA and GlyA bind to two opposite surfaces of FolD.

Lastly, since several proteins in the folate as well as 1-carbon metabolism pathway are homo-oligomers (as evidenced by the high fluorescence values along the diagonal), we asked if the surfaces mediating weak PPIs have the potential to disrupt the oligomeric interface. To that end, we looked at the homo-dimeric as well as the hetero-dimeric structures of GlyA and PurK, which show strong homo-oligomer signals as well as PPI with multiple proteins. Interestingly, in a few representative cases that we analyzed (FolA:GlyA in folate pathway; PurD:PurK in purine pathway; and FolA:PurK in the folate–purine inter-pathway), the interface mediating weak PPI was completely different from the oligomeric interface (Fig. EV8), indicating that these are truly novel interaction interfaces predicted by AlphaFold3 that do not rely on the sticky hydrophobic patches on the oligomeric interfaces.

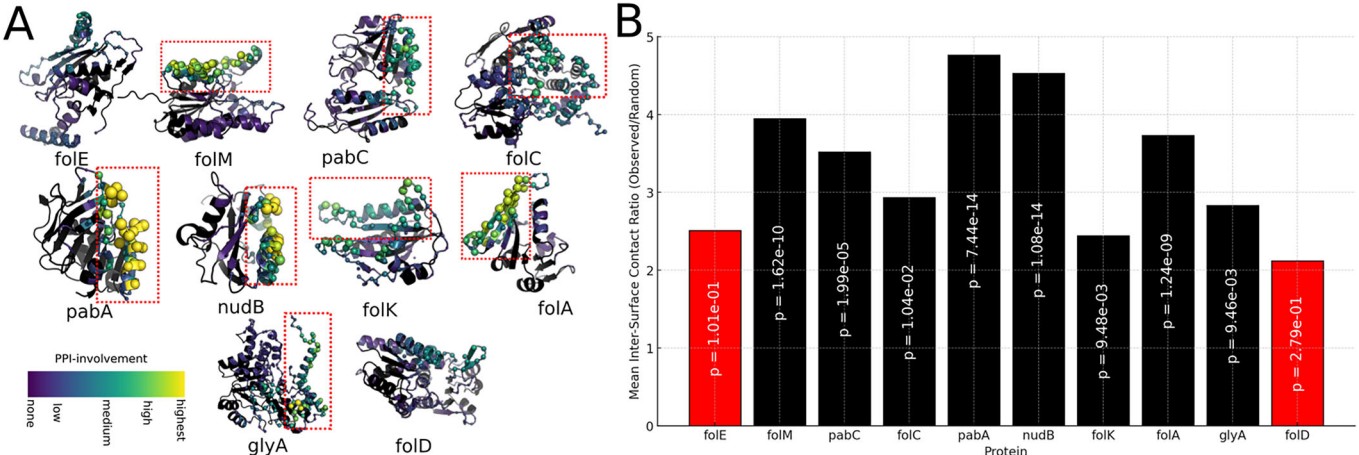

**Figure 7. Folate pathway proteins use dedicated conserved surfaces to interact with all their metabolon partners.**

(A) Representation of amino acid residues that are most frequently involved in heterodimeric protein–protein interactions in the folate pathway. Ten folate pathway enzymes are represented in the figure, and the most frequently observed residues in PPIs are represented by spheres. The radii and color of the spheres are represented such that most residues that appear most frequently in PPIs (in simulation) are represented by larger spheres that are brightly colored. Smaller, dark shaded spheres indicate residues that are not frequently involved in PPIs with other proteins. (B) The ratio of observed inter-surface contacts to inter-surface contacts expected under the null hypothesis for randomly occurring surfaces. This ratio is presented for all ten folate-pathway proteins. *p* values vs random overlap null model (see "Methods") is shown for each protein. Source data are available online for this figure.

## Folate-pathway metabolons show a dramatic increase in metabolic flux for the pathway compared to monomeric enzymes

Our experiments and simulations so far show the presence of PPIs between enzymes of the same pathway. To verify that interaction networks derived from fluorescence experiments result in the formation of spatial enzyme clusters, we employ a Coarse-Grained (GC) model where the enzymes are modeled as patchy particles. Each enzyme in the Langevin dynamics (LD) simulations is modeled using a central hard sphere of radius 10 Å and two patches on the surface of this hard sphere corresponding to a PPI-site and a substrate-binding site (Fig. 8A). The substrate-binding and PPI-site patches have ten distinct identities to account for the ten enzyme types of the folate pathway being modeled here (see Fig. 8A where we show the different coarse-grained patchy particle enzymes). Five independent trajectories of 10 μs each were used to study the functional dynamics of this multi-enzyme system. As seen from Fig. 8B, the enzymes form dense dynamic clusters that are stable at simulation timescales. Interestingly, even a relatively simple PPI energetics based on the broad MFI ranges discussed previously results in a dense PPI map observed in the simulations (Fig. 8C). The cluster size distribution at equilibrium (Fig. 8D) shows an exponential-like dependence and presence of several stable large clusters. These results show that a non-isotropic interaction model with a single interaction surface shared with multiple partners and with PPI energetics derived from the fluorescence experiments (Fig. 2) can support the formation of transient enzyme clusters within the folate pathway.

The results so far show that stable enzyme clusters could be supported for a heterogenous interaction map where only a subset of the enzyme pairs show stable interactions. How does the formation of dynamic clusters affect the net metabolic flux of the folate enzyme pathway? To address this question, we introduce a diffusion-reaction protocol in our LD simulations such that the ten patchy particle enzymes in the simulation catalyze ten sequential reactions in a linear reaction cascade (see Ranganathan et al (2023) and "Methods" section for more details).

In this study, we performed the diffusion-reaction simulations for two scenarios—

i) No PPIs are imposed, ensuring the enzymes are in a monomeric state, and all reactions occur in the bulk.
ii) Energetics of PPI between enzymes are derived from the experimental PPI matrix (Fig. 2) corresponding to the folate pathway (Fig. 8C).

Five independent 10 μs trajectories were simulated for both scenarios. The output of each reaction is tracked at the end of the simulation. In Fig. 8E, we plot the ratio of the number of successful reactions in the clustered state for enzymes normalized by the same for monomeric enzymes. Interestingly, as we move along the pathway (R4 and higher in Fig. 8E), the reaction fluxes show a significant gain for clustered enzymes, at the expense of a slight drop in flux for the initial reactions of the pathway. The gain in efficiency for later steps in the pathway can be several orders of magnitude compared with that for the monomeric unclustered state.

Our results, therefore, suggest that experimentally determined PPI between enzymes of the folate pathway results in enzyme clusters, which can improve the efficiency of metabolic turnover of the pathway. The reaction probabilities chosen in this study are drawn from a realistic distribution of $k_{cat}/K_M$ that peaks several orders of magnitude lower than the diffusion-limit Appendix Fig. S6). Therefore, our results strengthen the hypothesis that clustering of enzymes via PPIs can improve metabolic fluxes of pathways made up of otherwise "imperfect" enzymes [Fig. EV10 and (Ranganathan et al,

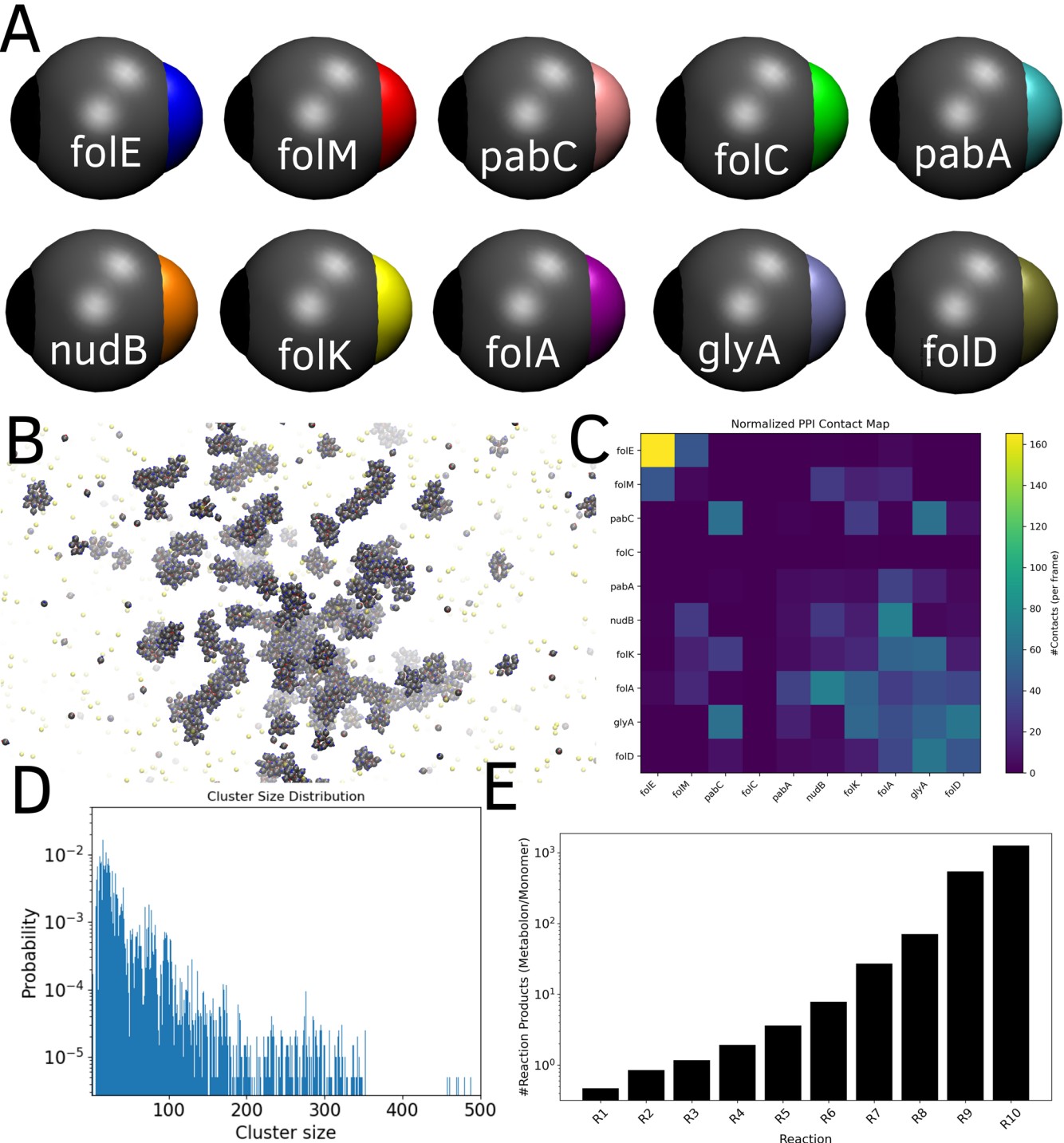

**Figure 8.    Reaction-diffusion Langevin dynamics simulations of the folate pathway.**

(**A**) Patchy particle model describing the ten folate pathway proteins. (**B**) Formation of enzyme clusters during the 1 μs Langevin dynamics simulations. (**C**) Mean numbers of protein–protein contacts between different coarse-grained enzyme particles in the Langevin dynamics simulations. (**D**) Size distribution of enzyme clusters at equilibrium. (**E**) Reaction fluxes for each reaction in the metabolon state normalized by the same quantity in the monomeric (unclustered) state of the enzymes in the control simulation where PPI are abrogated. The initial reactions in the pathway (R1–R4) show either a slight drop or no change in flux upon clustering. Later reactions in the pathway, however, show a dramatic increase in fluxes approaching several orders of magnitude showing the net gain in efficiency of substrate channeling in longer pathways upon clustering of enzymes. Source data are available online for this figure.

2023)], which operate far from the diffusion limit. Crucially, simulations with the patchy particle model show that even when the enzymes can interact via a single interface shared across several partners, the enzymes can exist in a clustered state that can support several orders of magnitude higher fluxes for the pathway compared to runs occurring in bulk solution. The stoichiometries of enzyme assemblies that form via the folate pathway PPI in our simulations (Appendix Fig. S7) reveal that all the enzymes of the pathway are enriched to the same extent in large clusters.

## Discussion

The one-carbon folate biosynthesis pathway comprising three intersecting and inter-dependent pathways (de novo purine, pyrimidine, and methionine synthesis) is one of the central metabolic networks of the living cell. It has long been hypothesized that metabolon formation may be essential to this pathway to protect labile folate co-factors that are easily prone to oxidative damage (Stover and Field, 2011). In the absence of reducing agents, folate derivatives were shown to have a half-life of a few minutes in vitro (Zheng and Cantley, 2019); however, they appear much longer-lived in in vivo studies (Krumdieck et al, 1978; von der Porten et al, 1992). Moreover, the fact that several mammalian proteins in the 1-carbon metabolism pathway had more than one binding site for folate co-factors indicated that substrate channeling was potentially inherent to this pathway (Schirch and Strong, 1989). In our earlier work, immunoprecipitation followed by mass spectrometry showed extensive albeit weak and transient PPIs between DHFR and its functional neighbors (Bhattacharyya et al, 2016). In this work, we systematically delineate the extent of PPIs in the entire *E. coli* 1-carbon metabolism pathway, using a bimolecular fluorescence complementation assay that can capture transient interactions in vivo. We identified two clusters of proteins in the folate and purine biosynthesis pathways, respectively, that had very strong intra-cluster as well as inter-cluster interactions.

A remarkable finding of our study was that the extent of interaction was the strongest among members of the folate biosynthesis pathway, followed by those in the purine biosynthesis pathway. Based on our data, there was no statistically significant interaction among the pyrimidine biosynthesis pathway proteins, though in several cases, they formed strong homomers (along the diagonal in Fig. 2A). This trend in interactions aligns quite strikingly with the order of stabilities of the metabolites in the respective pathways as folate metabolites (5,10-methelene THF, 5-methyl THF, N-formyl THF) are considered the most labile (Zheng and Cantley, 2019). On the other hand, pyrimidine pathway intermediates are mostly stable metabolites in the form of mono-phosphates or tri-phosphates (OMP, UMP, UTP, CTP, dTMP, etc.) and presumably do not need metabolon-like networks to protect them in the cytosol, while purine pathway intermediates are less stable (GAR, FGAR, AIR, NCAIR, SAICAR), as they are often difficult to quantify from the cell lysate using metabolomics (Bhattacharyya et al, 2021; Madrova et al, 2018). This observation may also hint more generally towards the possible evolution of PPI networks and metabolons in enzymatic networks to protect labile metabolites. Spatial clustering of functionally related enzymes into a dense cluster could also result in a significant increase in metabolic flux compared to the bulk reaction scenario, as previously demonstrated by mechanistic models (Ranganathan et al, 2023; Castellana et al, 2014).

Though much evidence is present in the literature about the existence of metabolons in various organisms, very little information is available about the structural architecture of these complexes (Bulutoglu et al, 2016; Skalidis et al, 2023; Wu and Minteer, 2015), particularly the nature of these PPI interfaces and their binding affinities. Some interesting questions are: do multiple proteins in a metabolon associate together, or do binary interactions occur at a time? How do these weak interaction interfaces differ from obligate PPI? Are the active sites of the enzymes predominantly involved in the interaction interface? A key highlight of the current study is that it provides structural insights into the structural detail of the 1-carbon metabolon using a combination of high-throughput mutagenesis and screening experiments, followed by metadynamics simulations in combination with AlphaFold3 prediction of initial complexes. Not only did the calculations predict PPIs with high accuracy as well as predict interfaces that agree with experimental findings, but they also shed light on some remarkable and unanswered aspects of the extended folate pathway metabolon. First, it shows that proteins that interact with multiple partners in the pathway almost always use the same interface for interaction with multiple metabolon partners, indicating a high degree of promiscuity at the interaction interface. This also suggests that monomeric enzymes are presumably involved in only binary interactions at a time. However, for oligomers, multiple subunits can associate with different proteins, and hence higher-order complexes may form, though we do not have such evidence so far from our present study. Second, our data on DHFR also shows that the PPI interface most used by the enzyme is far away from the active site. Proteins that interact with multiple partners (the so-called hub proteins) can be of two classes: date hubs (singlish hubs), wherein pairwise interactions happen one at a time, or party hubs (multi-hubs), which can engage multiple partners at the same time (Han et al, 2004; Peleg et al, 2014). Our findings suggest that promiscuous hub proteins in the 1-carbon metabolism pathway behave more like date hubs. Whether this is a general phenomenon for pathway enzymes warrants further investigation. For example, in the only other study where some degree of structural information is available for metabolons (Wu and Minteer, 2015), cross-linking followed by mass spectrometry showed that citrate synthase of the TCA cycle uses two distinct PPI interfaces to interact with malate dehydrogenase and aconitase, akin to FolD in our study. As observed in this study, the PPI interface was distinct from the active site. In the case of purinosomes (Deng et al, 2012), a luciferase reporter-based Tango reporter assay to assess PPI revealed that three enzymes, hPPAT, hTrifGART, and hFGAMS, of the purine biosynthesis pathway form a core structure, while other proteins associate with them. In the absence of any structural information, it is unknown if the enzymes use a common interface or can associate simultaneously; hence, future work will establish if our observations are a general property of metabolons.

Nevertheless, one might surmise as to why evolution might favor singlish hubs over multi-hubs, as date hubs ensure that one interaction must dissociate for another new interaction to happen, and therefore usage of the same PPI interface might be important for the dynamic assembly and disassembly of interactions that are so fundamental to the concept of metabolons. Shared interfaces might also pose a natural limit on the size of the metabolon, which otherwise can form giant assemblies that completely sequester proteins. It can also act as a

density modulator that can increase metabolic fluxes in long biochemical pathways (Ranganathan et al, 2023).

Our finding about the *E. coli* 1-carbon metabolon is one of the few examples of metabolons in prokaryotic cells. Much work in the recent past have shown that biochemical reactions do not take place in an unorganized manner in the cytosol of bacteria, rather there is a considerable degree of self-organization inside them. Other examples of such supramolecular enzyme clusters in bacteria include the TCA cycle pathway in *Bacillus subtilis* (Meyer et al, 2011) and the glycolysis pathway in *E. coli* (Mowbray and Moses, 1976).

Our previous work on *E. coli* DHFR (Bhattacharyya et al, 2016) suggested that one of the most important fitness consequences of metabolons is that when individual components are over-expressed, it sequesters others in the non-physiological complex forming near permanent interactions, thereby causing toxicity to the cells. A more detailed study on the propensity of a protein to be involved in PPI and its over-expression toxicity on a much larger dataset of proteins could shed light on sequestration of partner proteins through interaction as a general mechanism of toxicity in metabolic enzymes.

One might argue that our study merely shows interactions among multiple components of the pathway and did not present evidence of substrate channeling. Zhang and Fernie (2021) argue that only enzyme assemblies that are shown to both physically and functionally interact (channel metabolites) should be termed metabolons. However, considering that we observe the strongest interaction among enzymes with the most labile and short-lived substrates (folate pathway), and no interaction among the pyrimidine pathway with the most stable intermediates, suggest that substrate channeling must have been an evolutionary driving force to form transient complexes to protect unstable ligands from degradation by limiting their diffusion paths in the cytosol. Furthermore, our dynamic modeling of the folate pathway with and without formation of metabolon showed that the metabolon formed via experimentally observed PPI speeds up the flux through the pathway by several orders of magnitude due to substrate channeling. Therefore, the modeling results highlight the clear biological significance of the observed PPI in the folate pathway. This speed-up effect is especially pronounced for longer pathways, such as the folate pathway in this study, as the advantage from metabolon formation is less pronounced (and actually could be reversed in some cases) for shorter pathways comprised of 2–3 reactions (Ranganathan et al, 2023). Crucially, the increased fluxes for the pathway in our simulations were purely an outcome of PPIs without an explicit substrate channeling mechanism incorporated. PPIs could therefore have evolved to optimize biochemical yields for the whole pathway, while individual enzymes could operate far from the diffusion limit. A more explicit substrate channeling mechanism could further enhance the efficiency of enzyme clusters beyond that achieved by a simple clustering of enzymes via PPIs. Whether enzyme clusters that are much smaller than phase-separated droplets or condensates can sufficiently enhance reaction fluxes is a natural question that arises, considering the findings from our current study. The shape, size, stoichiometry, and dynamic properties that result as an outcome of shared PPI surfaces as opposed to multiple interaction patches would be critical to holistically understand the biological implications of the current work. Future work with more detailed insight into the effect of clustering on metabolic fluxes will decisively answer the outstanding questions concerning the functional and evolutionary significance of metabolons.

# Methods

### Reagents and tools table

| Reagent/resource | Reference or source | Identifier or catalog number |
|---|---|---|
| Western breeze chromogenic immuno-detection kit | Thermo Fisher Scientific | Cat # WB7105 |
| Pierce BCA protein assay kit | Thermo Fisher Scientific | Cat #23225 |
| NuPAGE Novex-4–12% Bis–Tris gels | Thermo Fisher Scientific | Cat # NP0321BOX |
| Ni-NTA Agarose | Qiagen | Cat #30210 |
| Trans-Blot Turbo Mini 0.2 μm Nitrocellulose Transfer Packs | BioRad | Cat #1704158 |
| **Experimental models** | | |
| *E. coli* BW27783 | CGSC (Coli Genetic Stock Center) | Cat No: 12119 |
| *E. coli* BL21(DE3) | Invitrogen | Cat No: C600003 |
| *E. coli* DH5-alpha | Invitrogen | Cat No: 18258012 |
| **Recombinant DNA** | | |
| pET28a vector | Novagen (Sigma) | Cat No: 69864 |
| **Antibodies** | | |
| Polyclonal anti-GFP antibody | Invitrogen | Cat # A11122 |
| Goat anti-rabbit ALP conjugated secondary Ab | Sigma-Aldrich | Cat # A3687 |
| **Oligonucleotides and other sequence-based reagents** | | |
| Primers used for cloning | IDT | Table EV4 |
| **Chemicals, enzymes, and other reagents** | | |
| BamHI, XbaI | NEB | |
| KOD DNA polymerase | Sigma | Cat No: 71085 |
| 10× Bugbuster reagent | Sigma | Cat No: 70921 |
| Complete EDTA-free protease inhibitor cocktail | Roche | Cat No: 11873580001 |
| Low melting agarose, OmniPur grade | Sigma | 2070-100GM |
| LB media | BD Biosciences | Cat No: 244620 |
| M9 medium | BD Biosciences | Cat No: 248510 |
| Bacto Casamino acids | Thermo Fisher Scientific | Cat No: 223050 |
| Glass bottom dishes | Willco Wells | Cat #HBST-3512 |
| Arabinose | Thermo Fisher Scientific | Cat No: 365180250 |
| IPTG | Thermo Fisher Scientific | Cat No: 15529019 |
| **Software** | | |
| ImageJ | https://imagej.nih.gov/ij/index.html | |
| GraphPad Prism | https://www.graphpad.com | |

| Reagent/resource | Reference or source | Identifier or catalog number |
|---|---|---|
| **Other** | | |
| Gel doc | In house | |
| PCR machine | In-house BioRad machine | |
| Trans Blot Turbo transfer system | BioRad | Cat # 1704150 |
| BD LSR Fortessa | Bauer core (Harvard) | |
| BD FACS Aria Cell Sorter | Bauer core (Harvard) | |
| Horiba XploRa confocal Raman microscope | Harvard core facility | |
| Zeiss Cell Observer microscope | Harvard Center for Biological Imaging | |

## Cloning and transformation of split YFP plasmids

The genes corresponding to 41 proteins in the folate pathway were amplified from the E. coli genome using primers containing BamHI and XbaI sites (Dataset EV4). The genes were cloned in two different formats in two mutually compatible plasmids: as a fusion to the N-terminus of YFP under the control of IPTG-inducible tac promoter, and as a fusion to the C-terminus of YFP under the control of an arabinose-inducible pBAD promoter. The YFP used in this work was the monomeric version of enhanced YFP (EYFP, truncated at 155/156) (Ohashi et al, 2012). This is brighter and matures faster than regular YFP, but still requires incubation at 30 °C for full maturation.

Out of the 41 proteins, 35 proteins were successfully cloned as fusions. NYFP fusion plasmids were transformed into BW27783 cells by the TSS method (Chung et al, 1989). This method enables transformation by simple incubation at 4 °C without any heat shock, and thereby allows high-throughput transformation in multi-well plates. Subsequently, cells transformed with NYFP-fusion plasmids were grown and made competent by the TSS method. For the NYFP-fusion plasmid, 35 CYFP-fusion plasmids were transformed, resulting in a 35 × 35 matrix of all possible combinations.

## Flow cytometry to detect interaction by split YFP assay

For FACS analysis, BW27783 cells transformed with both NYFP and CYFP fusion proteins were inoculated from glycerol stocks into 200 µl of supplemented M9 medium (containing Ampicillin and Chloramphenicol) in deep well plates and grown at 37 °C overnight. The next day, the cultures were diluted 1:100 in 150 µl of supplemented M9 medium containing antibiotics and inducers (0.2% arabinose and 1 mM IPTG), grown at 30 °C for 4–5 h, following which YFP fluorescence was detected by FACS. For analysis, the BD LSR Fortessa was used along with the HTS (high throughput) module. For each well of the 96-well plate, 30,000 cells were analyzed, and the mean fluorescence intensity was calculated. Those wells for which less than 10,000 cells could be analyzed were discarded.

## Western blot

To detect intracellular abundance of YFP fusion proteins, BW27783 cells transformed with a single plasmid (either NYFP or CYFP fusion proteins) were used. Cultures were inoculated from glycerol stocks into 200 µl of supplemented M9 medium (containing Ampicillin or Chloramphenicol, depending on the plasmid) in deep well plates and grown at 37 °C overnight. The next day, the cultures were diluted 1:100 in 5 ml supplemented M9 medium containing antibiotics and inducers (0.2% arabinose or 1 mM IPTG) in a 24-well deep well plate and grown at 30 °C for 4–5 h. The cultures were spun down, and lysis was carried out in 100 µl of Bugbuster solution in 1 × TBS for 30 min while shaking. Complete EDTA-free protease inhibitor cocktail from Roche was added during the lysis step. Subsequently, 100 µl lysate was mixed with 20 µl of 50% glycerol + 10% SDS solution and heated at 95 °C for 10 min. This dissolves the lysate and results in a clear solution. 15 µl of the cleared lysate was mixed with 5 µl of 4× loading dye, heated at 95 °C for 15 min, and loaded onto a 4–12% Bis–Tris gel. The leftover sample was used for the estimation of total cellular protein content using a BCA kit.

Western breeze chromogenic immuno-detection kit (Thermo) was used for western blot, following the manufacturer's guidelines. For detection of YFP fusion proteins, polyclonal anti-GFP antibody (Cat # A11122 from Invitrogen) was used as a primary antibody at a dilution of 1:2000, while goat anti-rabbit antibody conjugated to alkaline phosphatase was used as a secondary antibody. ImageJ was used to quantify the bands following blotting, and the intensities were normalized by total protein abundance obtained from the BCA method.

## Microscopy

BW27783 cells transformed with both NYFP and CYFP fusion proteins were inoculated from glycerol stocks into 2 ml of supplemented M9 medium (containing Ampicillin and Chloramphenicol) and grown at 37 °C overnight. The next day, the cultures were diluted 1:100 in 5 ml of supplemented M9 medium containing antibiotics and inducers (0.2% arabinose and 1 mM IPTG), grown at 30 °C for 4–5 h. For live phase contrast images, 2 µl of the culture was directly spotted on 1.5% low melting agarose (Calbiochem) pads. The agarose was dissolved in supplemented M9 medium. Pads were then flipped on a class #1.5 glass dish (Willco Wells), and the images were acquired at room temperature with a Zeiss Cell Observer microscope.

## Library generation

Based on the structure of DHFR (PDB: 7DFR), we selected all surface-exposed residues of DHFR (accessibility >80%), excluding its active site and those contacting active site residues within 5 Å radius. There were 85 such positions in DHFR that satisfied the above criteria, and at each of these positions, a VDS library was created. VDS codes for a combination of [ACG][AGT][GC] bases, which in total code for 1530 codons and 1105 amino acids. These codons code for all different amino acid types (hydrophobic, polar, charged); however, they do not incorporate large hydrophobics (F, W, or Y) or proline, and do not incorporate any stop codon. The mutagenesis was carried out on the NYFP-DHFR plasmid in a 96-well PCR plate using a megaprimer-based method, following which the reactions from individual wells were combined, digested overnight using DpnI to remove the WT plasmid, and transformed into DH5-alpha competent cells. A small amount was plated for single colonies, while the rest of the culture was grown overnight in the presence of antibiotics. The plasmid isolated from single clones was sequenced to confirm the presence of mutations. Following this, the library plasmid was isolated from the culture using miniprep.

## Sorting of clones

The NYFP-DHFR VDS library plasmid pool was transformed into the background of BW27783 cells containing CYFP-MetF/CYFP-FolK plasmids using a cell: plasmid ratio of 100:1 to minimize chances of multiple plasmids transformed into a single cell. The cultures were grown overnight at 37 °C, and the next day they were diluted 1:100 into fresh supplemented M9 medium containing antibiotics and inducers (0.2% arabinose and 1 mM IPTG). Using the BD FACS Aria Cell Sorter, single clones from the top and bottom 5% of the fluorescence intensity distribution of both cultures were sorted into wells of a 96-well plate that contained fresh supplemented M9 medium without any antibiotics. The single clones were allowed to grow for 3 days, following which they were diluted in medium containing antibiotics and allowed to grow overnight. The next day, after growth in the presence of inducers, fluorescence intensity was measured on the Fortessa analyzer, and those that had a substantial change in fluorescence compared to the WT culture were sent for sequencing to identify the mutation.

## Blinding

Data blinding was not relevant for this study; hence, no blinding was done.

## Permutation-based *p* value estimation for statistical significance of PPI clusters

To assess the statistical significance of observed interaction strengths within specified protein clusters, we employed a non-parametric permutation test. For each cluster, we computed a summary statistic (in this case, median) of the observed fluorescence values and compared it to a null distribution generated by randomly sampling an equal number of protein pairs from the full dataset. This process was repeated 10,000 times to build the empirical null distribution. The empirical *p* value was calculated as the proportion of null statistics greater than or equal to the observed value, using the following formula:

$$p = \frac{\#(S_{\text{null}} \geq S_{\text{obs}}) + 1}{N + 1}$$

where $S_{\text{obs}}$ is the observed statistic, $S_{\text{null}}$ are the null statistics, and $N$ is the number of permutations. This method of calculating *p* value does not require any assumption of normal distribution of the data.

## Raman spectroscopy

To better understand any plausible interaction between DHFR and FolK, we performed Raman spectroscopy. As a vibrational spectroscopy technique, Raman spectroscopy provides a host of structural information and hence has been in use for mechanistically probing conformational changes in proteins (Benevides et al, 2003; Rygula et al, 2013; Tuma, 2005). As compared to other spectroscopic techniques, Raman spectroscopy is essentially a non-destructive technique and is known for its high sensitivity. Raman spectroscopy is an informative technique in probing conformational changes upon binding interactions in proteins (Lippert et al, 1976). Conformational changes primarily impact the amide I range (1600–1700 $\text{cm}^{-1}$), which arises due to C = O stretching vibrations

of the peptide bonds, in turn modulated by the secondary structural elements, viz. helices, sheets, turns, etc. We probed into the secondary structure of DHFR and FolK using Raman spectroscopy with an aim to investigate the solution status after mixing. Any interaction upon mixing would essentially give rise to new spectral features, which, upon careful examination, help in the mechanistic understanding of conformational changes upon binding. Control experiments were run in the presence of ADK (detailed in the "Results" section), and protein concentration in all the acquisitions was kept at 15 µM.

The Raman spectroscopy studies utilized a Horiba XploRa confocal Raman microscope, featuring a thermoelectrically cooled detector at −70 °C and a 1200 gr/mm grating optimized for 750 nm. A 785 nm solid-state laser was employed as the excitation source, with each measurement lasting 180 s. To ensure accuracy, four spectra were gathered for each sample, facilitating cosmic ray removal. Spectral smoothing was achieved using the Horiba denoise algorithm, and a polynomial fit was applied for fluorescent baseline correction prior to peak deconvolution in Labspec 6. The 785 nm laser's power was maintained at ~41 mW to avoid photo-bleaching or thermal damage, given the excitation wavelength's significant separation from protein absorption bands. Settings included a 200 µm spectrometer slit, a 500 µm confocal aperture, and calibration against a 520.7 $\text{cm}^{-1}$ silicon reference. Experiments used 20 µM protein solutions in sodium phosphate buffer, with a 20 µl volume, alongside buffer blanks for thorough comparison. Spectral data were normalized against the distinct 330 $\text{cm}^{-1}$ peak.

### Protocol for generating random surfaces

To ensure that the difference in inter-surface contacts for the observed and randomly generated surfaces is not merely due to differences in the surface sizes, we followed the following protocol while generating random surfaces.

For each of the ten folate pathway proteins under study, we compute the radius of gyration of the residues that make up the "*n*" interaction surfaces for each protein. The mean radius of gyration for the surface is inferred from these values. This is used as the radius of an idealized spherical interaction surface.

i) To draw the random surfaces, we define a list of solvent accessible residues for every folate protein based on DSSP calculations. All residues with a relative solvent-exposed surface area >0.25 were considered exposed and were considered while defining random surfaces. For every random surface, we pick a residue from this exposed residue list and draw a probe radius (computed in step (i)) around it to define a set of exposed residues that make up the random surface.

ii) To account for the irregularities in protein surfaces and the accompanying variability in the number of $C_\alpha$ atoms that make up each surface, we employ a normalized contact number to compute statistical differences. The raw contacts are normalized by $m_i * m_j$ where $m_i$ and $m_j$ refer to the number of $C_\alpha$ atoms that are part of the *i*th and *j*th random surfaces (see Eq. 2).

## Metadynamics simulations

Thirty-five enzymes corresponding to the Folate, Purine, and Pyrimidine pathways were used in the computational study. These

correspond to the same set of genes listed in the experimental PPI matrix shown in Fig. 2. We use AlphaFold 2.0 multimer build to predict the pairwise dimeric structures for all intra-pathway pairs within the three pathways. We also randomly sample 55 dimers, each corresponding to the Folate–Purine and Purine–Pyrimidine interactions, to estimate inter-pathway PPIs.

To account for dynamicity in interactions and compute the stability of the AlphaFold predicted dimers, we employed metadynamics simulations wherein the distance between two chains in the dimer is used as the collective variable of interest (Fig. 5A). We then compute the free energy landscape as a function of these inter-chain distances for every folate dimer under study (Fig. 5A). Minima at shorter distances (<interaction radii of the pair) corresponds to a favorable dimer. We then integrated over all bound and unbound structures in the landscape to compute the binding free energy ($\Delta G_{\text{binding}}$ in Fig. 5B). All simulations were performed using NAMD2.3 (Phillips et al, 2020) with the CHARMM36 forcefield. Interaction cutoffs of 12 Å and 13.5 Å were set for van der Waals and electrostatic interactions. A simulation timestep of 2 fs was used for the study. Metadynamics simulations were run till the PMFs exhibited convergence.

## Statistical test to check for non-random overlap of interaction surfaces

To check if there is a non-random overlap of interaction surfaces across different interaction partners, we define an order parameter —inter-surface $C_\alpha$–$C_\alpha$ contacts as a measure of shared interaction surfaces. To compute this quantity, we define for each folate protein an interaction surface—a list of interacting residues—with every other protein that it interacts with in the folate pathway. To draw this residue list, we use structures that populate the free energy minima in our metadynamics simulations or revealed by high-throughput mutagenesis experiment. For instance, in meta-dynamics simulations FolA shows stable interactions with four other partners. Therefore, FolA has four potentially distinct interaction surfaces for each of these different partners. We then compute $C_\alpha$–$C_\alpha$ contacts for the set of FolA residues that define each of the four surfaces (across all possible surface pairs) and then compute a quantity—MISC$^O$—Mean Inter-surface Contacts derived from actual PPI surfaces obtained from mutational mapping or metadynamics simulations—for each of the ten folate proteins.

$$\text{MISC}^O = \sum_{i=1}^{n} \sum_{j=i+1}^{n} \frac{C^O(S_i, S_j)}{n(n-1)/2} \qquad (1)$$

Here superscript O refers to observed (as opposed to random control) surfaces. $n$ is the number of detected interaction surfaces for the protein. $S_i$ and $S_j$ represent the list of residues that make up the $i$ and $j$th surfaces, respectively. The function $C^O(S_i, S_j)$ represents the normalized number of pairwise $C_\alpha$–$C_\alpha$ contacts between residues that make up the $i$ and $j$th surfaces.

$$C^O(S_i, S_j) = \frac{\sum_{s=1}^{m_i} \sum_{p=1}^{m_j} \Delta\left(\left|\vec{r}_s^{\,i} - \vec{r}_p^{\,j}\right|\right)}{m_i m_j} \qquad (2)$$

Here, $m_i$ is the number of residues belonging to $i$th surface, $\{\vec{r}_s^{\,i}\}$ represents the set of coordinates of all $C_\alpha$ atoms belonging to

surface $i$ and

$$\Delta\left(\left|\vec{r}_s^{\,i} - \vec{r}_p^{\,j}\right|\right) = \begin{cases} 1 \text{ if } \left|\vec{r}_s^{\,i} - \vec{r}_p^{\,j}\right| \leq 7\text{Å} \\ 0 \text{ if } \left|\vec{r}_s^{\,i} - \vec{r}_p^{\,j}\right| > 7\text{Å} \end{cases}$$

is a contact function that defines contact between $C_\alpha$ atoms belonging to different surfaces with 7 Å cutoff.

If the interaction surfaces for different partners exactly overlap, the mean inter-surface contacts (observed) would be high. On the other hand, if these surfaces were distinct and non-overlapping, this quantity would be very small. We compute these mean inter-surface contacts (observed) for each of the ten enzymes of the folate pathway.

To determine the statistical significance of the overlap between different interacting surfaces, we propose a null hypothesis that the interaction surfaces are randomly distributed on the surface of a protein in question. In other words, we test whether the mean inter-surface contacts (observed) are significantly higher than the analogous quantity MISC$^R$—Mean Inter-surface Contacts (Random)—for a set of 100 randomly drawn surfaces for every folate protein under study.

$$\text{MISC}^R = \sum_{i=1}^{n_R} \sum_{j=i+1}^{n_R} \frac{C^R(S_i, S_j)}{n_R(n_R - 1)/2} \qquad (3)$$

Where $n_R = 50$ the number of randomly generated surfaces The function $C^R(S_i, S_j)$ represents the normalized number of pair-wise $C_\alpha$–$C_\alpha$ contacts between the $i$ and $j$th randomly generated surfaces, defined analogously to Eq. 2 but using $C_\alpha$ coordinates of randomly generated surfaces (see below for the algorithm that generated random surfaces). The mean inter-surface contacts (random) are then compared to the inter-surface contacts observed in experiment and simulations to assess the non-random nature of their overlap. Ten thousand such random sets are generated, and the distribution of the inter-surface contacts in the random control is then computed (Fig. EV9). Our null hypothesis is that there is no significant difference between the observed mean normalized contacts and what would be expected by random chance. In other words, any observed difference is due to random variation per the null hypothesis.

To determine the significance of the observed test statistic, we perform a $t$ test for statistical significance. Here, the $Z$ score is calculated using the following equation,

$$Z = \frac{\text{MISC}^O - \text{MISC}^R}{D} \qquad (4)$$

Here, the denominator $D$ refers to the variation in the inter-surface contacts, for the randomly generated surfaces.

$$D = \sigma/\sqrt{n} \qquad (5)$$

Here, $\sigma$ refers to the standard deviation in the normalized pair-wise contact number (Eq. 2) for the randomly generated surfaces, whereas "$n$" refers to the number of randomly generated sets used for the computation of the statistic. The $Z$ score measures how many standard deviations from the mean the observed value lies and

determines $p$ value for the underlying $t$-statistic for the random null model. We set a significance level (e.g., 0.05), which represents the threshold for statistical significance. If the $p$ value is less than this significance level, we reject the null hypothesis. Otherwise, the null hypothesis is accepted.

## LD simulations

### *The Patchy Particle Enzyme Model*

To study how PPIs between enzymes modulate reaction fluxes in the folate biosynthetic pathway, we employed off-lattice LD simulations implemented in the LAMMPS (Thompson et al, 2022) molecular dynamics engine. This approach captures both thermal motion and viscous damping of particles, allowing us to simulate the coupled effects of diffusion, interaction, and reaction in a biologically realistic environment.

Patchy hard-sphere models are widely used to study the self-assembly of multivalent proteins (Espinosa et al, 2020; Tejedor et al, 2021). In our framework, each enzyme is represented as a rigid hard sphere bearing two adhesive patches (Fig. 8A) that mimic—(i) catalytic active site and (ii) interface for protein protein interaction analogous to the interfaces in Fig. 7. The catalytic patches carry distinct "identities," ensuring that only one specific ligand type can bind to each active site. Similarly, the PPI interface patch also has ten distinct identities corresponding to ten different enzymes in the Folate pathway.

Substrate molecules themselves are modeled as diffusing hard spheres, which react only upon meeting their complementary patch. In Fig. 8A, the enzyme core is shown as a larger gray sphere, while its active-site patches and PPI patches are shown as smaller colored hemispheres on the surface of the large spheres. Although substrate molecules may transiently associate anywhere on the enzyme surface via these non-specific forces, a catalytic event occurs only when a ligand contacts its matching active-site patch.

We model a sequential enzymatic pathway consisting of $N$ turnover steps (see Eq. 6 and Fig. 1A). In this scheme, each enzyme catalyzes a transformation in the sequence

$$S_0 \rightarrow S_1 \rightarrow S_2 \rightarrow ... \rightarrow S_9 \rightarrow P \qquad (6)$$

so that, after $N$ consecutive reactions, the initial substrate $S_0$ is converted into the final product P within an enzyme cluster or in the bulk.

In these simulations, the enzyme particles are surrounded by a pool of substrates that are entirely in an $S_0$ state corresponding to the first reaction in the pathway. To capture both diffusion and catalysis, we implement a stochastic diffusion-reaction algorithm: each time a substrate or intermediate $S_i$ occupies its cognate active-site patch on enzyme $E_i$, it may proceed to $S_{i+1}$ with probability $P_{\text{react}}$ during that timestep. Whenever a substrate or intermediate binds to its matching active-site patch, it may convert to the next chemical state with probability $P_{\text{react}}$.

Each active-site patch supports two classes of binding:

a.) Cognate binding (strength $\varepsilon_{\text{co}}$) occurs only between an active-site patch and its specific ligand, ensuring that each enzyme catalyzes a defined step in the linear pathway (Eq. 1). The strength of cognate binding in our simulations is $5k_{\text{B}}T$.

b.) Non-cognate binding (strength $\varepsilon_{\text{nc}} = \varepsilon_{\text{PL}}$) represents all other, non-specific associations between patches and off-target substrates or intermediates. The strength of non-cognate binding in our simulations is $0.8k_{\text{B}}T$. In our model, $\varepsilon_{\text{nc}}$ is set equal to the generic enzyme–ligand interaction $\varepsilon_{\text{PL}}$, so that any non-complementary patch–ligand pair interacts solely via the same weak, non-specific forces governing the enzyme surface and ligand molecules.

## Potential functions and simulation parameters

All non-specific interactions—enzyme–enzyme interactions outside the PPI patch ($\varepsilon_{\text{PP}}$), enzyme–ligand outside the active site ($\varepsilon_{\text{PL}}$), and non-cognate patch–ligand contacts ($\varepsilon_{\text{nc}}$)—are described by a standard Lennard–Jones potential.

$$U_{LJ}(r) = 4\varepsilon \left[ \left(\frac{\sigma}{r}\right)^{12} - \left(\frac{\sigma}{r}\right)^6 \right] \qquad (7)$$

Where $\sigma$ is the sum of the two particle radii ($\sigma_{\text{enz}} = 20$ Å for enzyme cores; $\sigma_{\text{patch}} = \sigma_{\text{ligand}} = 8$ Å for patches and ligands), and the cutoff distance $r_{\text{c}} = 2.5\sigma$. The depth $\varepsilon$ of the attractive well is chosen to set the desired interaction strength—$\varepsilon_{\text{PP}}$ for protein–protein, $\varepsilon_{\text{PL}}$ for non-specific protein–ligand, and $\varepsilon_{\text{nc}}$ for non-cognate binding are set to $0.8\,kT$.

Specific (cognate) binding between (a) an active-site patch and its matching ligand, and (b) PPI patches on enzymes is implemented via a Morse potential such that,

$$U_{\text{Morse}}(r) = D_e \left[ e^{-2\alpha(r-r_0)} - 2e^{-\alpha(r-r_0)} \right] \qquad (8)$$

Here, $D_e$ controls the strength of the attractive interactions, $r_0$ is the equilibrium distance for the interaction and $\alpha$ is a parameter that controls the sharpness of the potential. For PPIs, the interaction patches on enzymes experience short-range attractive interactions, with $\alpha = 2.2$ Å$^{-1}$, and $r_0 = 17.056$ Å. The cutoff distance was set to 20 Å. The range of the attractive interaction between PPI patches in our simulations is set to 2.8 Å, enforcing single-valent binding per patch. The strength of interaction between any two PPI patches is modeled based on the fluorescence intensities of the corresponding enzyme pair in experiments (Fig. 2A). The $K_D$ values for all interacting enzyme pairs with MFI > 1000 are set to 10 μM. For MFI between 800 and 1000, the $K_D$ for the interaction is set to 200 μM. For MFI between 650 and 800, the $K_D$ is set to 500 μM. All MFIs <650 are modeled with a $K_D$ of 100 mM. These $K_D$s are translated into interaction strengths between PPI patches in the Morse potential such that $D_e = K_B T.\ln(K_D)$. For the sake of computational simplicity, we assume that the entropic contribution to the free energy is negligible.

This interaction facilitates transient enzyme clustering, enhancing local concentration. Similarly, cognate binding of substrate onto the complementary active site on enzymes is modeled via the Morse potential with $\alpha = 1$ Å$^{-1}$, and $r_0 = 12.7$ Å. The range of the attractive interaction between substrate and a cognate binding site is 2.5 Å, ensuring that only one substrate molecule can bind to an active site at any given time. For cognate binding, $D_e = \varepsilon_{\text{co}} = 5$ kT.

The equations of motion for each rigid patchy particle is

$$m\frac{d^2r}{dt^2} = F_c + F_f + F_r \qquad (9)$$

where $F_c = -$delta_U is the sum of conservative forces from Eqs. 7 and 8.

$F_f = -\alpha \, v$ is the viscous drag, and $F_r$ is a stochastic force with

$$\langle F_r(t) \rangle = 0 \text{ and } \langle F_r(t)F_r(t') \rangle = 2\alpha k_B T.\delta(t - t').$$

Simulations were carried out in LAMMPS under NVT conditions using a Langevin thermostat at $T = 300$ K and solvent viscosity $\eta = 10^{-3}$ Pa·s.

The damping coefficient $\alpha$ governs how rapidly momentum is relaxed and is related to solvent viscosity and particle radius by:

$$\alpha = \frac{K_B T}{6\pi\eta a} \tag{10}$$

We employed a timestep $\Delta t = 30$ fs. Because each enzyme is treated as a rigid multi-center body (no internal degrees of freedom), we used LAMMPS's rigid particle integrator to enforce fixed relative positions of the core and its patches.

The simulation box, in our study, contains 300 copies of each of the 10 coarse-grained enzyme types such that the effective concentrations of enzymes in the simulation box is 50 μM. In order to facilitate collisions between enzyme molecules at the simulation timescale, during the first 1 μs of the simulation the enzyme molecules are confined within a spherical region of 400 Å radius in the center of the cubic simulation box. The spherical confinement is then relaxed, and the enzyme clusters are allowed to equilibrate during the rest of the simulation run.

### Parameterization of reaction probabilities

To model enzymatic turnover in a coarse-grained simulation of a ten-step metabolic pathway, we derived per-attempt reaction probabilities for each enzymatic step based on the known distribution of $k_{cat}/K_M$ values observed in natural enzymes. A total of 10 representative $k_{cat}/K_M$ values were randomly sampled from a log-normal distribution with parameters derived from the enzyme $k_{cat}/K_M$ distribution presented in Appendix Fig. S6 (based on data by Bar-Even et al (2015)), which spans approximately seven orders of magnitude ($10^2 - 10^9$ M$^{-1}$s$^{-1}$). We used a normal distribution in log-space with a mean of 5 and standard deviation of 1.0 to generate $\log_{10}(k_{cat}/K_M)$ values. These were exponentiated to yield real-world $k_{cat}/K_M$ values. The value of $P_{react}$ for the 10 reactions is chosen based on a known distribution of $k_{cat}/K_M$ values (normalized by the $k_{cat}/K_M$ for the diffusion limit $\rightarrow 10^8$ M$^{-1}$s$^{-1}$) (see "Methods" and Appendix Fig. S6). Each sampled value was then normalized by the diffusion limit, taken to be $10^8$ M$^{-1}$s$^{-1}$, which reflects the maximum catalytic efficiency constrained by molecular diffusion in the cytoplasm. The resulting normalized reaction efficiencies, $\eta_{react} = (k_{cat}/K_M)/10^8$ represent the relative catalytic efficiency of each enzyme compared to an idealized diffusion-limited enzyme. Note that $\eta_{react}$ is a dimensionless quantity which we then use to set values of $P_{react}$ for the ten reactions in our diffusion-reaction model (see Appendix Fig. S6).

## Data availability

This study does not include any data deposited in external repositories.

The source data of this paper are collected in the following database record: biostudies:S-SCDT-10_1038-S44320-025-00139-9.

## Peer review information

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

## Acknowledgements

This work was supported by NIH grant R35GM139571 to EIS. The authors thank the flow cytometry facility (Bauer core) and the Harvard Center for Biological Imaging core at Harvard University.

## Author contributions

**Sanchari Bhattacharyya**: Conceptualization; Resources; Data curation; Formal analysis; Supervision; Validation; Investigation; Visualization; Methodology; Writing—original draft; Project administration; Writing—review and editing. **Srivastav Ranganathan**: Resources; Data curation; Software; Formal analysis; Validation; Investigation; Visualization; Methodology; Writing—original draft; Writing—review and editing. **Sourav Chowdhury**: Data curation; Formal analysis; Validation; Investigation; Visualization; Methodology; Writing—original draft; Writing—review and editing. **Bharat V Adkar**: Resources; Data curation; Software; Formal analysis; Validation; Investigation; Visualization; Methodology. **Mark Khrapko**: Investigation. **Eugene I Shakhnovich**: Conceptualization; Formal analysis; Supervision; Funding acquisition; Writing—original draft; Project administration; Writing—review and editing.

Source data underlying figure panels in this paper may have individual authorship assigned. Where available, figure panel/source data authorship is listed in the following database record: biostudies:S-SCDT-10_1038-S44320-025-00139-9.

## Disclosure and competing interests statement

The authors declare no competing interests. The work was entirely performed when the authors were part of Harvard University.

# Expanded View Figures

**Figure EV1.  The original asymmetric matrix shows similar cluster formation as the symmetric matrix in Fig 2.**

(**A**) The complete asymmetric matrix of all interactions measured in both configurations (NYFP-A/CYFP-B) and (NYFP-B/CYFP-A). Figure 2A is derived from this matrix by taking the maximum value for each PPI pair. As in Fig. 2A, red boxes indicate MFI > 1100, yellow for 900 < MFI < 1100, blue for 750 < MFI < 900 while white boxes indicate MFI < 750. Gray boxes indicate no measurement. (**B**) Based on the original matrix in (**A**) and Dataset EV2, we calculate the statistical significance of the clusters identified in Fig. 2E. The results are largely similar to Fig. 2E. For each cluster, the red data point represents the median MFI value of the cluster, while the box represents a null distribution generated from the median values of an equal number of interaction pairs (as the cluster size) randomly picked 10,000 times from the matrix. The box represents 25–75 percentile of the data and line in between represents the median of the distribution. The whiskers represent the 5–95 percentile interval. Statistical significance and $p$ value are calculated using a one-sided non-parametric permutation test as described in "Methods." *** indicates $p$ value < 0.001, ** indicates $p$ value < 0.01. No data normality is assumed. $p$ value = 0.0002 for folate cluster, $p$ value = 0.0001 for purine pathway, $p$ value = 0.0001 for purine cluster, $p$ value = 0.0028 for folate–purine intercluster.

▶

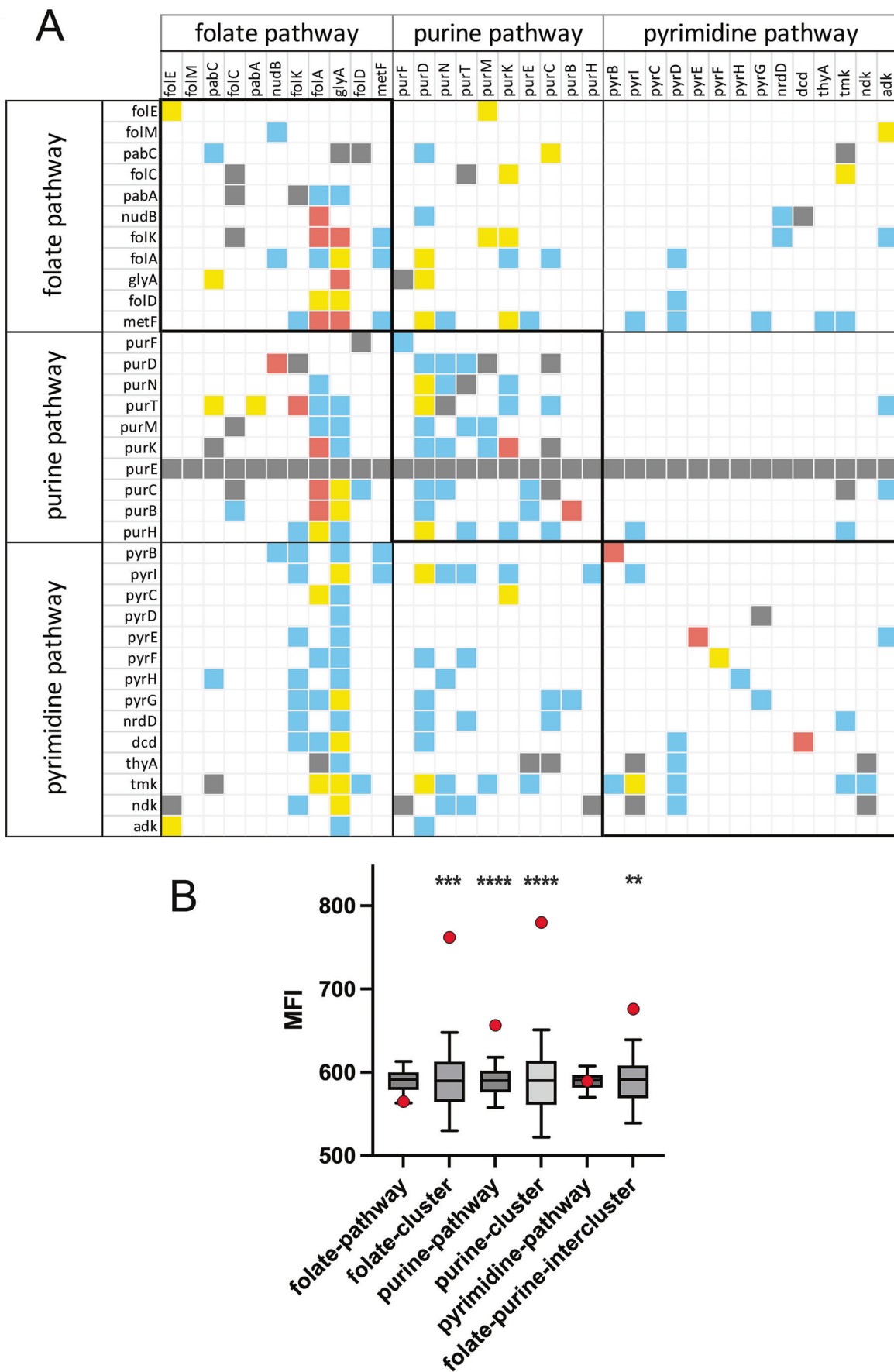

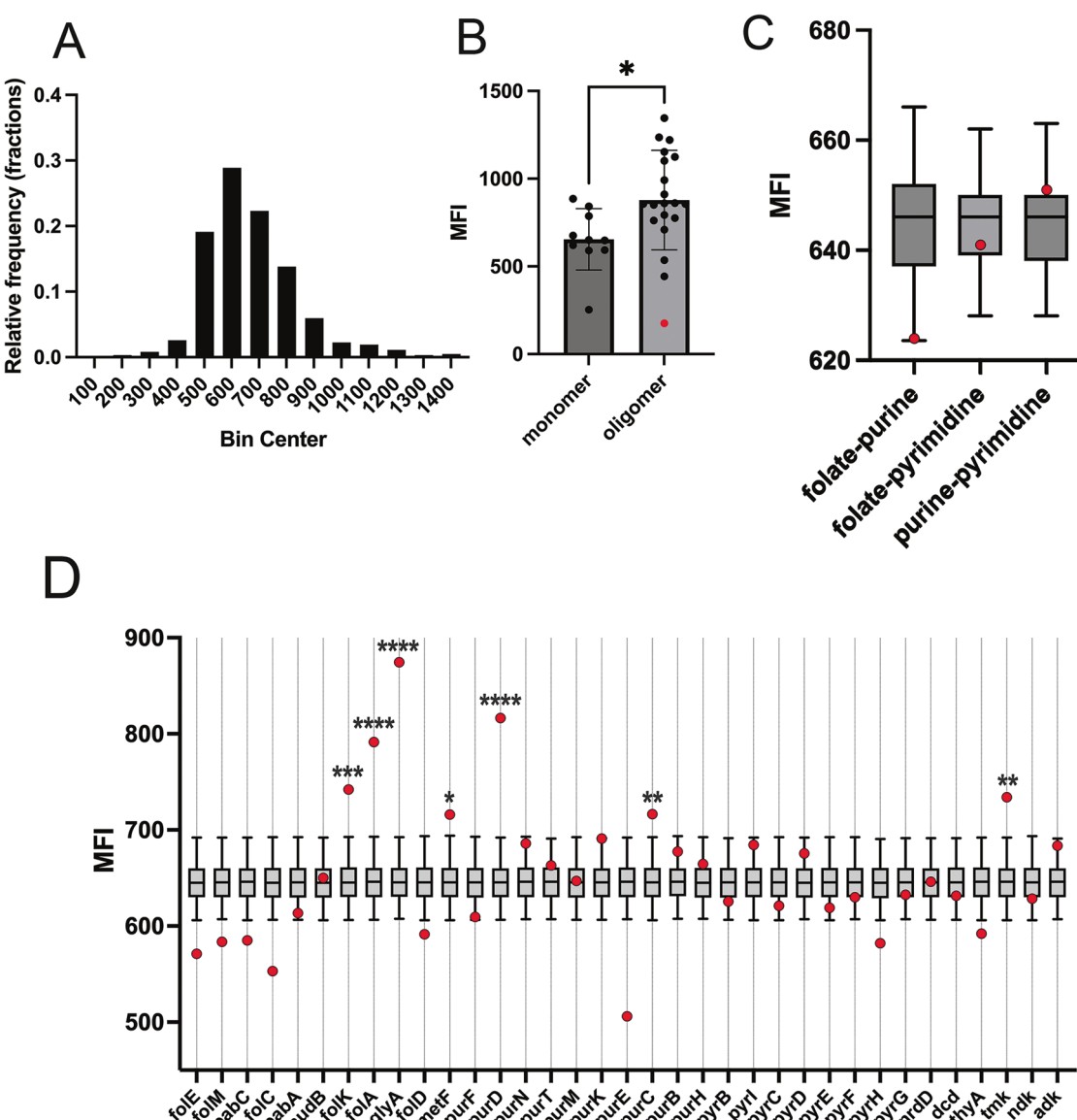

**Figure EV2. Cluster formation and interaction propensity of different proteins in the dataset.**

(A) Frequency distribution of mean YFP fluorescence intensities of 1225 interactions from the dataset. The intensities show a wide range of variation. (B) All proteins in our dataset for which oligomeric status is known from crystal structure data (total 30 proteins) were grouped into two classes—monomer and oligomer (dimer and above). The plot shows the distribution of MFI values (homo-oligomer interactions along the diagonal in Fig. 2E) for all proteins belonging to that group. Error bars represent standard deviation. * indicates $p$ value < 0.05, ** indicates $p$ value < 0.01. A two-tailed unpaired $t$ test shows that oligomers have significantly higher fluorescence intensity compared to monomers ($p$ value = 0.03, $n = 10$ for monomer set, and $n = 20$ for oligomer set). The red data point corresponds to PurT, for which the MFI of 175 was much lower than the cellular background fluorescence for non-interacting protein pairs. If this data point is excluded, then the $p$ value becomes strongly significant with $p$ value of 0.005. (C) There is no significant inter-pathway interaction among folate, purine, and pyrimidine biosynthesis pathways when entire pathways are taken into consideration. For each cluster, the median MFI value of the cluster (red data point) is compared against a null distribution that is generated from the median values of an equal number of interaction pairs (as the cluster size) randomly picked 10,000 times from the matrix (represented by the box plot, where the box represents 25–75 percentile of the data and line in between represents the median of the distribution. The whiskers represent the 5–95 percentile interval). Statistical significance and $p$ value are calculated using a one-sided non-parametric permutation test as described in "Methods." No data normality is assumed. (D) Interaction propensity of each protein with all other proteins in the dataset, represented as the average of MFI across all 35 proteins in the dataset. The median (red data point) of all 34 interactions of a protein (except with itself) is compared against a null distribution, as explained in (C). Cases for which the $p$ values were significant are shown with *. Overall, **** indicates $p$ value < 0.0001, *** indicates $p$ value < 0.001, ** indicates $p$ value < 0.01, while * indicates $p$ value < 0.05. The actual $p$ values were the following: FolK—0.0008, FolA—0.0001, GlyA— 0.0001, MetF—0.01, PurD—0.0001, PurC—0.0076, Tmk—0.0022. Statistical significance and $p$ value are calculated using a one-sided non-parametric permutation test as described in "Methods." No data normality is assumed. For box-plots, the box represents 25–75 percentile of the data and line in between represents the median of the distribution. The whiskers represent the 5–95 percentile interval.

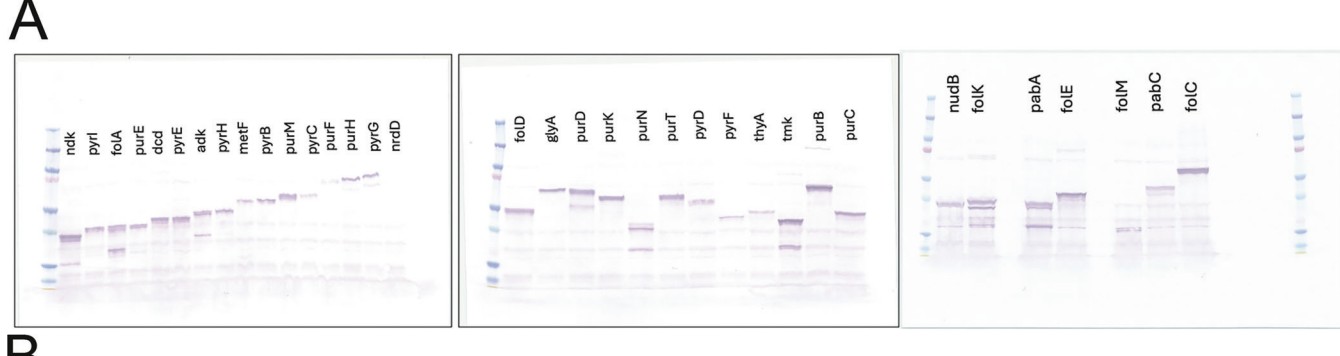

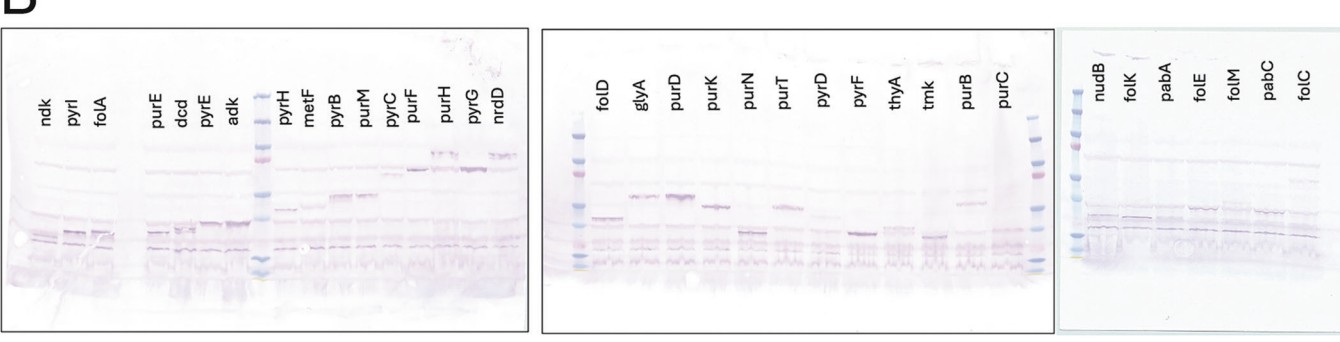

**Figure EV3.    Expression check for different NYFP and CYFP fusion constructs.**

Western blot images to check the expression of (**A**) NYFP- and (**B**) CYFP- fusion constructs using polyclonal anti-YFP antibody (see "Methods" for details). Since the NYFP fragment is larger, NYFP-fusion proteins have higher intensities on the blot than their CYFP counterparts, and therefore, blot (**B**) shows significantly more background. In all cases, the most prominent band that matches the expected molecular weight of the fusion proteins was used for quantification. Overall, though expression levels of the fusion proteins vary, they are not sufficient to explain differences in observed PPI strengths.

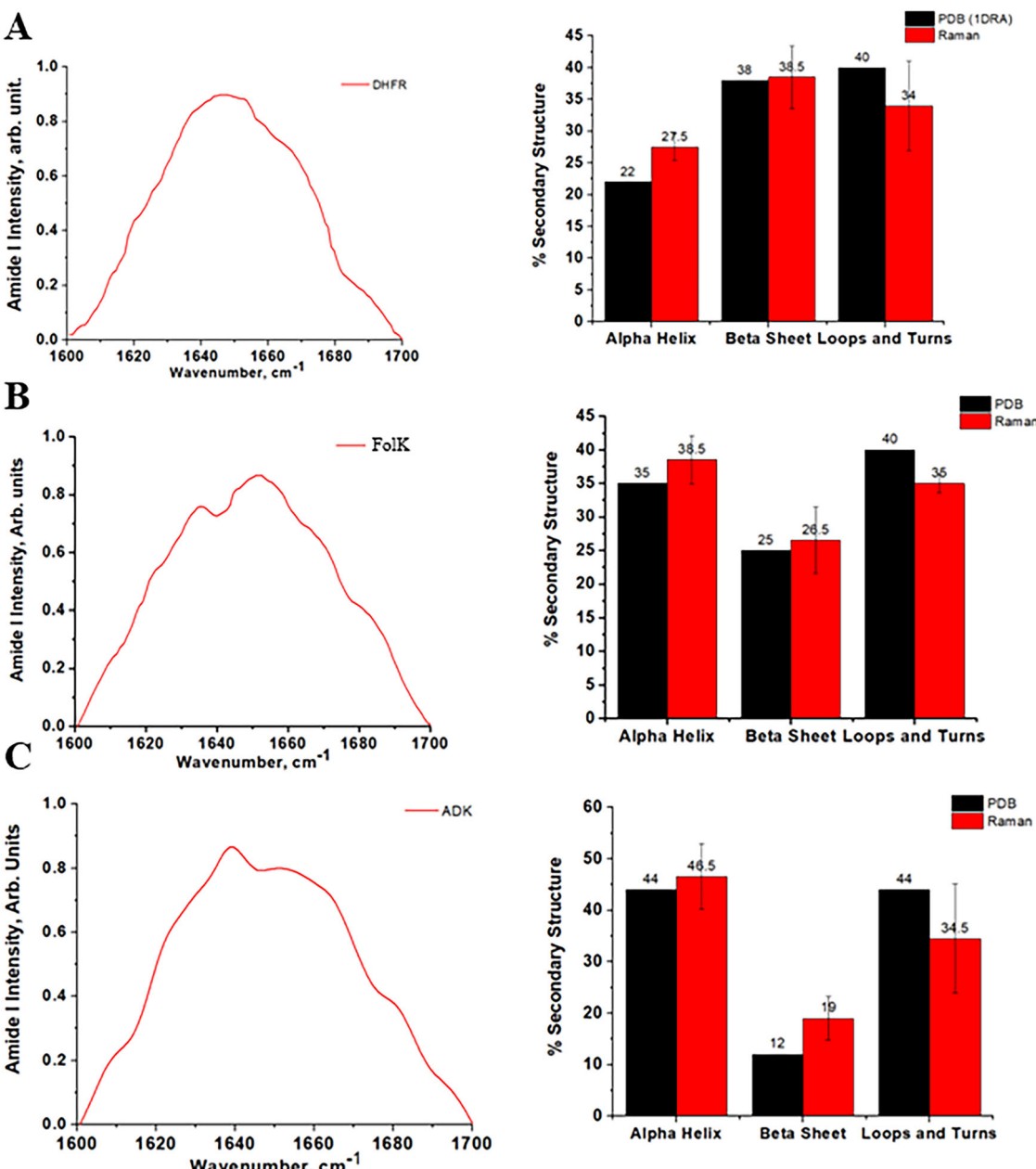

**Figure EV4. Comparison of secondary structure content from Raman spectra and the PDB structure.**

(**A**) Amide I (1600–1700 cm$^{-1}$) Raman spectral profile of WT DHFR. The plot shown is the mean of the acquisition from three independent technical repeats. The bar plot shows the secondary structure contents of DHFR (black bars) as obtained upon deconvolution of the Amide I spectrum. Amide I spectra for DHFR were deconvoluted using Lorentzian peak fitting and the secondary structural contents for helix, beta sheet, turns, and loops were compared with the solved structure (black bars, PDB 1DRA). Peaks were selected based on the standard Amide peaks used for secondary structure analysis, and the quality of the fit was checked using the reduced chi-squared value, which was typically between 1.2 and 1.5. The secondary structural content was found to be comparable to the PDB structure, further reflecting the quality of the spectra and the peak fitting carried out. (**B**) A similar analysis was done for FolK. The secondary structural content was found to be comparable to the PDB structure. (**C**) Amide I spectral profile of the control protein ADK was acquired. A similar analysis was done for ADK, and the secondary structural content was found to be comparable to the PDB structure. $n = 3$ biologically independent samples were used for the experiments. All the data are presented as mean values $+/-$ SEM.

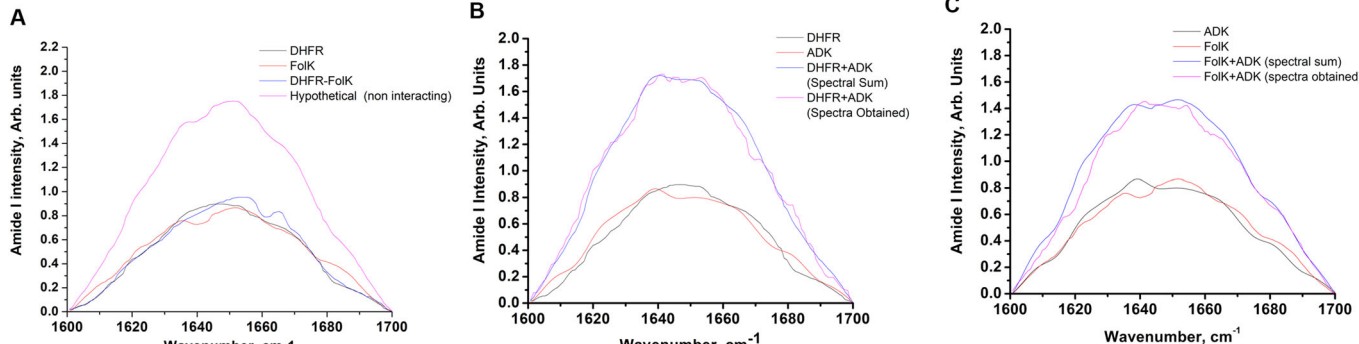

**Figure EV5.   Raman spectra of DHFR, ADK and FolK proteins alone and in complex.**

(**A**) Raman spectra obtained from DHFR–FolK interaction set do not overlap with the hypothetical spectrum, which is the mathematical sum of DHFR and FolK spectra. Raman spectra for control sets using ADK and each of DHFR (**B**) and FolK (**C**) were acquired using laser excitation of 785 nm. Intensities (arbitrary units, Arb. Units.) for native DHFR and ADK in the amide I range (1600-1700 cm$^{-1}$) were recorded individually and then upon mixing. The resultant spectra obtained were found to be comparable to the mathematical sum of the individual spectrum of DHFR and ADK, potentially suggesting the absence of any interaction. (**C**) The resultant spectrum was found to be comparable to the mathematical sum of the individual spectrum for ADK and FolK, which suggests the absence of any interaction. $n = 3$ biologically independent samples were used for the experiments. All the data are presented as mean values $+/-$ SEM.

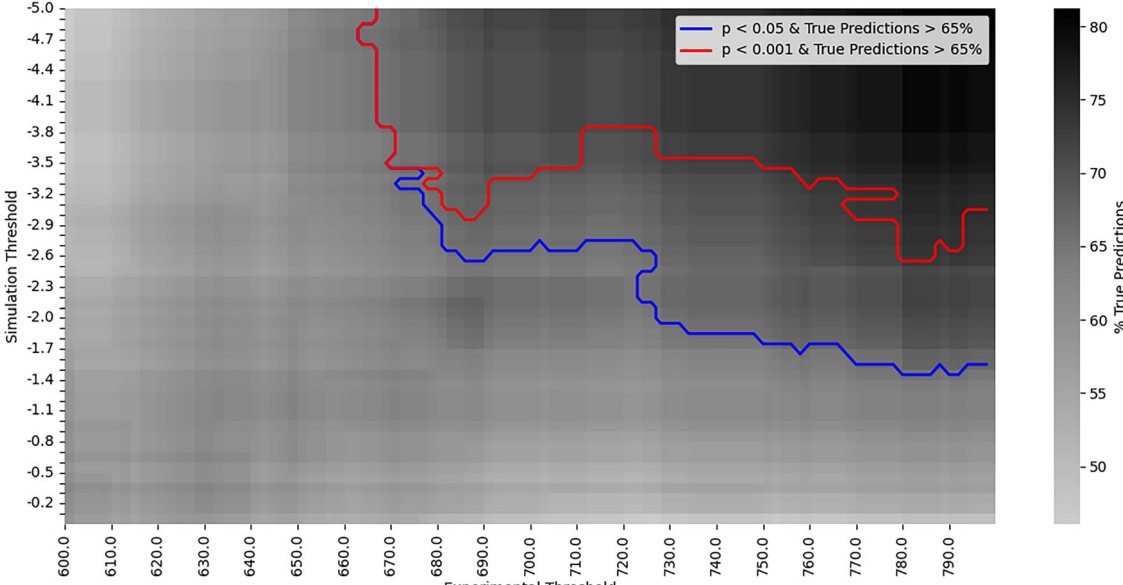

**Figure EV6.   Robustness of simulation predictions.**

The sensitivity of true predictions % according to simulation to varying thresholds of $\Delta G_{binding}$) and the experimental threshold value from fluorescence experiments (*x*-axis). The regions within the blue and red contours are statistically significant at *p* values of <0.05 and <0.001, respectively. The combination of thresholds within the shaded regions results in statistically significant and strong agreement, wherein >65% of stable binding predictions from simulations match that of experiments.

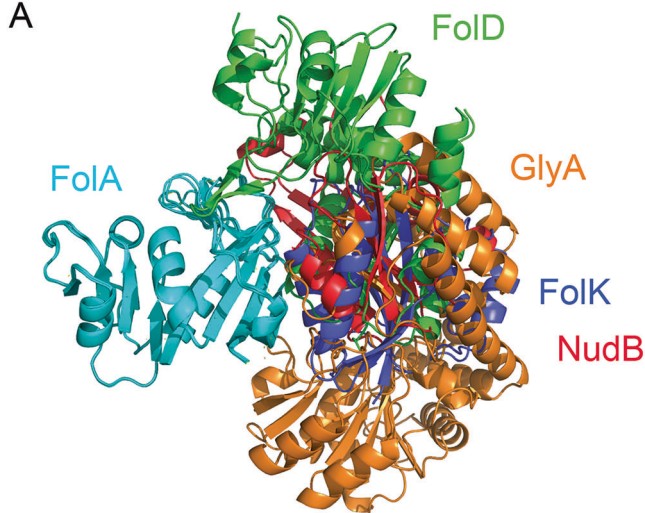

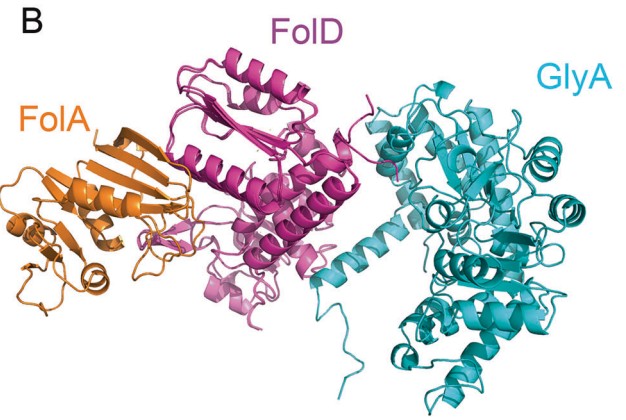

**Figure EV7.   Analysis of complex structures predicted by Alphafold.**

(A) Structural alignment of PPI complexes of FolA with different binding partners (FolD, GlyA, FolK, and NudB) shows that FolA uses a similar interface to interact with multiple proteins, hence the highly significant *p* value of FolA (*p* value = 1.2e−9 in Fig. 7B). (B) Structural alignment of PPI complexes of FolD with two different binding partners (FolA and GlyA) shows that the two interfaces are completely different, hence the non-significant *p* value (*p* value = 0.279 in Fig. 7B).

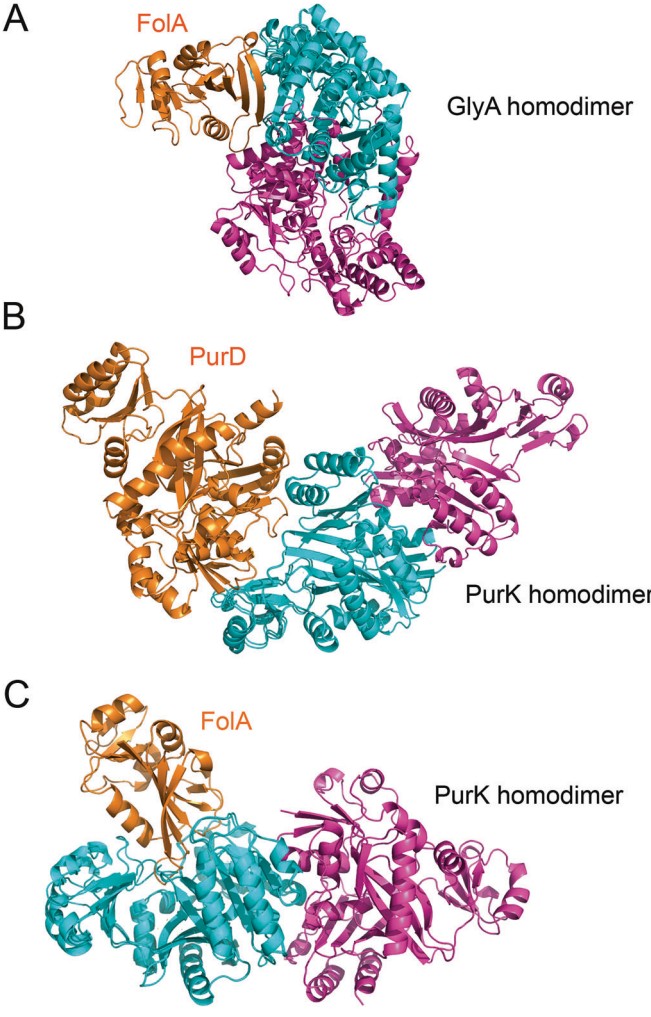

**Figure EV8. Overlay of representative heterodimeric structures of PPI complexes with their corresponding homodimeric structures.**

(A) FolA-GlyA complex superimposed with GlyA homodimer, (B) PurD-PurK complex superimposed with FolK homodimer, and (C) FolA-PurK complex superimposed with FolK homodimer. In all these structures, the homodimeric interface is different than the PPI interface.

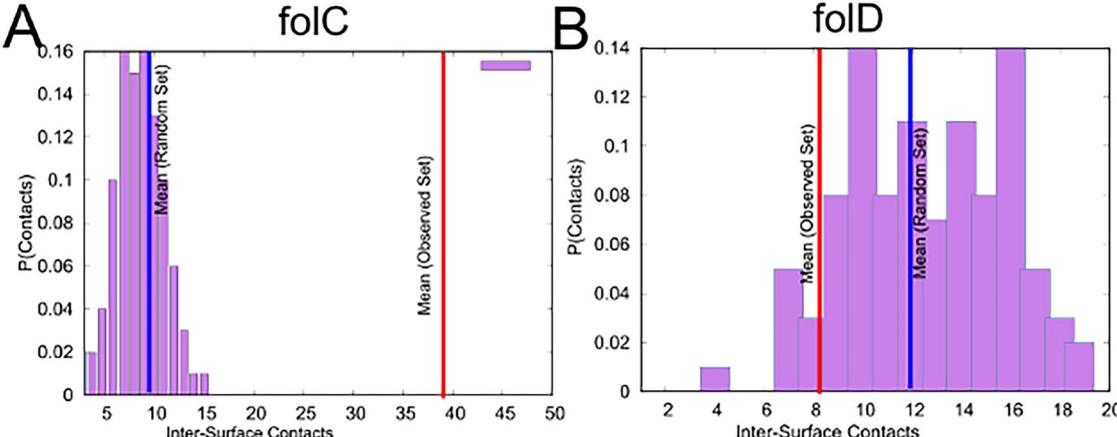

**Figure EV9. Proteins use similar interface to interact with multiple other proteins in the pathway.**

Distribution of mean pairwise inter-surface contacts for a set of 50 randomly drawn surfaces on (A) FolC and (B) FolD proteins. The distribution corresponds to the mean inter-surface contacts for 10,000 such realizations, each with 50 randomly drawn surfaces. The blue horizontal line shows the mean value for inter-surface contacts for the random set, while the red line shows the corresponding value for the observed surfaces. (A) For the FolC protein, the mean inter-surface contacts (observed) is significantly higher than the corresponding value for the random set, suggesting that the surfaces used for interaction across different proteins share a high degree of overlap. (B) On the other hand, the interaction surfaces on FolD corresponding to different interaction partners show no such overlap.

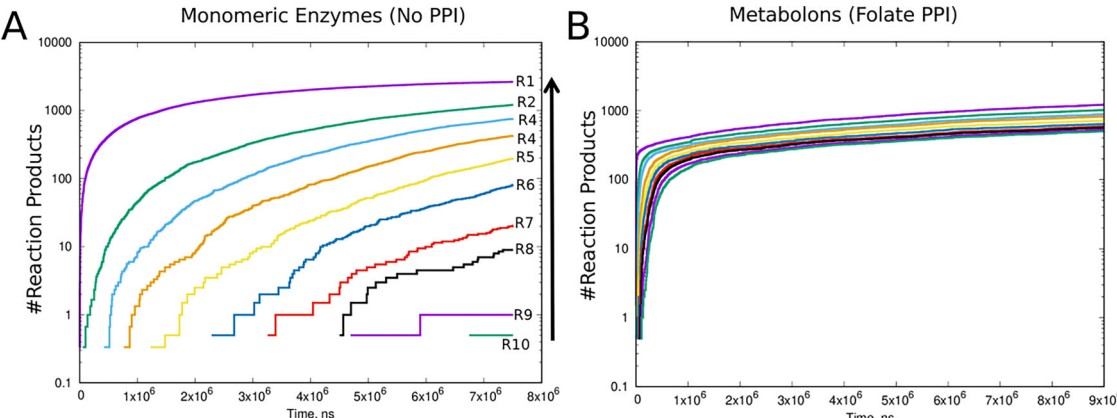

**Figure EV10. Protein-protein interactions significantly accelerate reactions compared to a diffusion controlled model.**

Progression of the ten reactions in the diffusion-reaction model during simulation for (**A**) monomeric enzymes where all reactions occur in the bulk and (**B**) when enzymes interact via a PPI map based on experiments.

