## [Peer Review File · Molecular Systems Biology]

Conserved interfaces mediate multiple protein-protein interactions in a prokaryotic metabolon

Sanchari Bhattacharyya, Srivastav Ranganathan, Sourav Chowdhury, Bharat Adkar, Mark Khrapko, and Eugene Shakhnovich

Corresponding author(s): Eugene Shakhnovich (shakhnovich@chemistry.harvard.edu) , Sanchari Bhattacharyya (sbhattacharyya@fas.harvard.edu)

Review Timeline:

Submission Date:	18th Jan 25
Editorial Decision:	1st Apr 25
Revision Received:	13th Jun 25
Editorial Decision:	22nd Jul 25
Revision Received:	10th Aug 25
Accepted:	14th Aug 25

Editor: Poonam Bheda

Transaction Report:

1st Apr 2025

Manuscript Number: MSB-2025-12868

Title: Conserved interfaces mediate multiple protein-protein interactions in a prokaryotic metabolon

Dear Prof Shakhnovich,

Thank you for the submission of your manuscript to Molecular Systems Biology. We have now received feedback from the three reviewers who agreed to evaluate your manuscript. As you will see from the reports below, the referees acknowledge the interest of the study and are overall supportive of your work; however they also comment on multiple aspects of the manuscript that should be strengthened in a revision.

I think that the recommendations of the reviewers are rather clear and I therefore do not see the need to repeat the comments listed below. All issues raised would need to be satisfactorily addressed. Please let me know in case you would like to discuss in further detail any of the any of the reviewer comments or your proposed revisions, I would be happy to schedule a call.

We require:

1) A .docx formatted version of the manuscript text (including legends for main figures, EV figures and tables). Please make sure that the changes are highlighted to be clearly visible. Alternatively you may choose to submit your manuscript as a LaTeX file.

4) A .docx formatted letter INCLUDING the reviewers' reports and your detailed point-by-point responses to their comments. As part of the EMBO Press transparent editorial process, the point-by-point response is part of the Peer Review File (PRF), which will be published alongside your paper.

5) A complete author checklist, which you can download from our author guidelines (<https://www.embopress.org/page/journal/17574684/authorguide#submissionofrevisions>). Please insert information in the checklist that is also reflected in the manuscript. The completed author checklist will also be part of the PRF.

6) Please note that all corresponding authors are required to supply an ORCID ID for their name upon submission of a revised manuscript.

7) It is mandatory to include a 'Data Availability' section after the Materials and Methods. Before submitting your revision, primary datasets produced in this study need to be deposited in an appropriate public database, and the accession numbers and database listed under 'Data Availability'. Please remember to provide a reviewer password if the datasets are not yet public (see <https://www.embopress.org/page/journal/17574684/authorguide#dataavailability>).

In case you have no data that requires deposition in a public database, please state so in this section as follows: "This study includes no data deposited in external repositories". Note that the Data Availability Section is restricted to new primary data that are part of this study.

8) All Materials and Methods need to be described in the main text using our 'Structured Methods' format, which is required for all research articles. According to this format, the Methods section includes a Reagents and Tools Table (listing key reagents, experimental models, software and relevant equipment and including their sources and relevant identifiers) followed by a Methods and Protocols section describing the methods using a step-by-step protocol format. The aim is to facilitate adoption of the methodologies across labs. Please upload the Reagents and Tools table as a separate document when submitting your revised manuscript. More information on how to adhere to this format as well as a downloadable template (.docx) for the Reagents and Tools Table can be found in our author guidelines:

<https://www.embopress.org/page/journal/17444292/authorguide#structuredmethods>

9) For data quantification: please specify the name of the statistical test used to generate error bars and p-values, the number (n) of independent experiments (specify technical or biological replicates) underlying each data point and the test used to calculate p-values in each figure legend. The figure legends should contain a basic description of n, p-values and the test applied. Graphs must include a description of the bars and the error bars (s.d., s.e.m.). Please provide exact p-values (in either the figure or figure legend).

10) Our journal encourages inclusion of *data citations in the reference list* to directly cite datasets that were re-used and obtained from public databases. Data citations in the article text are distinct from normal bibliographical citations and should directly link to the database records from which the data can be accessed. In the main text, data citations are formatted as follows: "Data ref: Smith et al, 2001" or "Data ref: NCBI Sequence Read Archive PRJNA342805, 2017". In the Reference list, data citations must be labeled with "[DATASET]". A data reference must provide the database name, accession number/identifiers and a resolvable link to the landing page from which the data can be accessed at the end of the reference. Further instructions are available at .

11) We replaced Supplementary Information with Expanded View (EV) Figures and Tables that are collapsible/expandable online. EV Figures should be cited as 'Figure EV1, Figure EV2' etc... in the text and their respective legends should be included in the main text after the legends of regular figures.

- Additional Tables/Datasets should be labeled and referred to as Table EV1, Dataset EV1, etc. Legends should be provided in a separate tab in case of .xls files. Alternatively, the legend can be supplied as a separate text file (README) and zipped together with the Table/Dataset file.

<https://www.embopress.org/page/journal/17574684/authorguide#expandedview>

12) Author contributions: CRediT has replaced the traditional author contributions section because it offers a systematic machine-readable author contributions format that allows for more effective research assessment. Please remove the Authors Contributions from the manuscript and use the free text boxes beneath each contributing author's name in our system to add specific details on the author's contribution. More information is available in our guide to authors.

13) Disclosure statement and competing interests: We updated our journal's competing interests policy in January 2022 and request authors to consider both actual and perceived competing interests. Please review the policy <https://www.embopress.org/competing-interests> and update your competing interests if necessary.

14) Every published paper now includes a 'Synopsis' to further enhance discoverability. Synopses are displayed on the journal webpage and are freely accessible to all readers. They include a short stand first (maximum of 300 characters, including space) as well as 2-5 one-sentences bullet points that summarizes the paper. Please write the bullet points to summarize the key NEW findings. They should be designed to be complementary to the abstract - i.e. not repeat the same text. We encourage inclusion of key acronyms and quantitative information (maximum of 30 words / bullet point). Please use the passive voice. Please attach these in a separate file or send them by email, we will incorporate them accordingly.

Please note that these would be the final versions and changes during proofing are usually not allowed.

15) As part of the EMBO Publications transparent editorial process initiative (see our policy here:

https://www.embopress.org/transparent-process#Review_Process), Molecular Systems Biology will publish online a Peer Review File (PRF) to accompany accepted manuscripts.

In the event of acceptance, this file will be published in conjunction with your paper and will include the anonymous referee reports, your point-by-point response and all pertinent correspondence relating to the manuscript. Let us know whether you agree with the publication of the PRF and as here, if you want to remove or not any figures from it prior to publication.

Please note that the Author checklist will be published at the end of the PRF.

Molecular Systems Biology has a "scooping protection" policy, whereby similar findings that are published by others during review or revision are not a criterion for rejection. Should you decide to submit a revised version, I do ask that you get in touch after three months if you have not completed it, to update us on the status.

Yours sincerely,

Poonam Bheda

Reviewer #1:

Thanks for asking me to review this outstanding manuscript where the authors address the formation of metabolons in the cell using highly innovative experimental and computational methods and introducing new concepts in biochemistry. Purine and pyrimidine biosynthesis uses folate as 1-carbon donor and the two pathways must be somehow coordinated by the cell. On these bases, the authors first performed an extensive analysis of possible protein-protein interactions among the enzymes of the folate and nucleotide pathways. They employed split YFP methodology. Having demonstrated the existence of interactions, the experimentally interrogated them by combining mutagenesis with a sophisticated usage of Raman spectroscopy. They could thereby see that enzymes tend to use the same surface for interactions with different proteins. These findings were then corroborated by the structural analysis using AlphaFold3. The conclusions of this work are of great significance: metabolons do form, they rely on partly promiscuous yet specific interacting surfaces that are away from the active site, and the propensity to metabolon formation inversely correlates with the the stability of the intermediate metabolites.

This is an excellent paper, one of the best (of the many) that I have been reviewing in the past times. I only have a few suggestions:

-The general reader will appreciate a more extensive explanation of the data shown in panels 1D and 1E.

-The legend of Figure 2A (the most critical one for the paper) is somewhat confusing. "While there is no significant clustering for all 11 proteins in the folate pathway (55 pairs of interactions), a set of 4 proteins.....", I have the impression that there are typing errors with regard to the numbers and names of the enzymes because the legend does not match the data shown in the panel and the main text narrative.

-I really do not understand why homo-oligomers should be detected by the split YFP method (diagonal elements in panel 2A and associated text). As shown in Figure 1B, YFP's fluorescence is reconstituted only when the N- and C-fragments come together due to the physical proximity of their respective carrier enzymes. Or can you have YN-YN and YC-YC pairs that give rise to a fluorescent species?

Reviewer #2:

Referee Report on "Conserved Interfaces Mediate Multiple Protein-Protein Interactions in a Prokaryotic Metabolon"

The manuscript presents an extensive mapping of protein-protein interactions (PPI) in the E. coli 1-carbon metabolism pathway, using a combination of high-throughput fluorescence assays, mutagenesis, and computational modeling. While the dataset is impressive and the findings intriguing, the biological significance of the reported interactions remains unclear. Below, I outline specific concerns that should be addressed to strengthen the manuscript.

Major Concerns:

1. Limited Biological Interpretation of Intra-Pathway Interaction Enhancement

The authors report a statistically significant enhancement of intra-pathway interactions (Fig. 3E), but the effect size appears modest. It is unclear whether this enhancement is functionally meaningful or merely a byproduct of the experimental method. Could the authors provide additional evidence or theoretical justification for why this level of enhancement would impact metabolic flux or enzyme function?

2. Translation of MFI Values into Biophysical Parameters

The manuscript heavily relies on mean fluorescence intensity (MFI) values to quantify interactions. However, MFI is not a direct biophysical measurement, making it difficult to interpret the binding dynamics. The authors should attempt to translate these values into more conventional biophysical terms, such as dissociation constants (K_d) or, ideally, the fraction of time each enzyme spends bound to its interaction partner(s). Without this, it is difficult to assess whether these interactions are transient or functionally relevant.

3. Lack of Biological Rationale for Conserved Interaction Surfaces

A key observation is that proteins in the metabolon use a common interface to interact with all their partners, resulting in exclusively dimeric interactions. While this is an interesting structural feature, its biological significance is not well explained. Why would evolution favor this particular interaction mode? Would it constrain or facilitate dynamic assembly and disassembly of the metabolon? A mechanistic model explaining the functional consequences of this interaction topology would strengthen the study.

Minor Issues:

4. Comparisons with Other Metabolons

The authors suggest that the observed interaction architecture might be a general feature of metabolons. Providing a comparison with known metabolon structures (e.g., purinosomes) would help contextualize these findings.

In conclusion, while the manuscript presents a valuable dataset and some novel insights, the biological significance of the

findings is not sufficiently clear. Addressing the concerns outlined above, particularly by providing a more quantitative analysis of binding strengths and a mechanistic rationale for the observed interaction architecture, would greatly improve the impact of the study.

Reviewer #3:

The authors utilized biomolecular fluorescence complementation (BiFc) to extensively map weak protein-protein interactions (PPIs) in the E.coli 1-carbon metabolism pathway. Thirty-five enzymes are fused to the N- and C-YFP, and the complete 1225 pairs are analyzed using flow cytometry. Discovered interactions are further assessed using various techniques. They demonstrated certain conserved region interacts with multiple partners. Though the systematical map of PPIs using BiFc may have some background noise from non-specific interactions, it provides a high throughput approach to uncover potential PPIs among intra- and inter- pathway enzymes. Researchers who interest in PPIs among multiple proteins may find this study informative.

Major points

1. The authors describe, "for each pair, fluorescence was measured in two configurations: NYFP-X/CYFP-Y and NYFP-Y/CYFP-X, where X and Y are the two proteins in question" and all the pairs were experimentally analyzed. However, "Fig 2A was generated by selecting the highest fluorescence value obtained in any of the configuration". It's known and understandable that the two configurations could generate different fluorescence values. But it's misleading to just show the highest value. The authors need to present the entire dataset from the two configurations in the matrix and use that to re-do the analysis presented in Figure 2E.

2. The authors claim that "we found that the increase in inducer concentration causes a sharp increase in YFP fluorescence (Fig 2B, C and D), indicating that the observed fluorescence is not non-specific, rather it arises from specific protein-protein interactions that are strongly concentration dependent." Both non-specific and specific PPIs can lead to YFP reconstitution, the protein expression dependent of YFP fluorescence cannot prove the interactions are specific.

3. The authors say, "Further the resultant spectrum was not the mathematical sum of individual spectra obtained with DHFR and FolK before interaction." However, the mathematical sum spectrum of DHFR and FolK is missing in the Figure 5A. The solely appearance of "new features around 1663 cm^{-1} " is not sufficient to convince others.

4. AF3 multimer modeling rely on a random seed; it is possible that different seeds will yield different predicted structures, yielding an ensemble of structure rather than a unique prediction. Is this the case for the AF models studied in this manuscript? In other words, how representative are the structures used for metadynamics simulations with respect to the ensemble of AF3 predictions?

Minor point

1. The split point for YFP used in this study is at 155/156, which has been shown with moderate self-assemble probability. Reference "Ohashi, K., Kiuchi, T., Shoji, K., Sampei, K., & Mizuno, K. (2012). Visualization of cofilin-actin and Ras-Raf interactions by bimolecular fluorescence complementation assays using a new pair of split Venus fragments. *Biotechniques*, 52(1), 45-50." Just want to bring this to authors attention and I don't expect any experiment about this matter.

Reviewer #1:

Thanks for asking me to review this outstanding manuscript where the authors address the formation of metabolons in the cell using highly innovative experimental and computational methods and introducing new concepts in biochemistry. Purine and pyrimidine biosynthesis uses folate as 1-carbon donor, and the two pathways must be somehow coordinated by the cell. On these bases, the authors first performed an extensive analysis of possible protein-protein interactions among the enzymes of the folate and nucleotide pathways. They employed split YFP methodology. Having demonstrated the existence of interactions, the experimentally interrogated them by combining mutagenesis with a sophisticated usage of Raman spectroscopy. They could thereby see that enzymes tend to use the same surface for interactions with different proteins. These findings were then corroborated by the structural analysis using AlphaFold3. The conclusions of this work are of great significance: metabolons do form, they rely on partly promiscuous yet specific interacting surfaces that are away from the active site, and the propensity to metabolon formation inversely correlates with the the stability of the intermediate metabolites.

This is an excellent paper, one of the best (of the many) that I have been reviewing in the past times. I only have a few suggestions:

We thank the reviewer for the appreciative comments.

-The general reader will appreciate a more extensive explanation of the data shown in panels 1D and 1E.

We have now included a more detailed explanation of the flow cytometry data in the legend to Figure 1 (Lines 610-618).

-The legend of Figure 2A (the most critical one for the paper) is somewhat confusing. "While there is no significant clustering for all 11 proteins in the folate pathway (55 pairs of interactions), a set of 4 proteins.....", I have the impression that there are typing errors with regard to the numbers and names of the enzymes because the legend does not match the data shown in the panel and the main text narrative.

As we understand, the reviewer here refers to Fig 2E, which analyses presence of clusters and infers their statistical significance. Fig 2A on the other hand is simply a color-coded representation of flow cytometry values, which signifies the extent of interaction among the proteins. Basically, a set of 5 proteins in the folate pathway (FolK, FolA, GlyA, FldD and MetF) and 5 proteins in the purine biosynthesis pathway (PurK, PurD, PurM, PurD, PurN) show significantly higher fluorescence than a random set of interactions from the matrix, indicating that they might form a cluster. No such evidence of cluster formation is evident for members of the pyrimidine biosynthesis pathway.

-I really do not understand why homo-oligomers should be detected by the split YFP method (diagonal elements in panel 2A and associated text). As shown in Figure 1B, YFP's fluorescence is reconstituted only when the N- and C-fragments come together due to the physical proximity of their respective carrier enzymes. Or can you have YN-YN and YC-YC pairs that give rise to a

fluorescent species?

In our study, gene corresponding to every protein in the selected pathways is cloned in 2 formats, NYFP-A and CYFP-A. Each NYFP fusion protein is co-transformed with each of the 35 CYFP fusion proteins, resulting in the 35x35 matrix. As a result, data points along the diagonal represent a scenario where plasmids coding for NYFP and CYFP fusions of the same protein (NYFP-A and CYFP-A) are co-transformed. In this case, we will observe NYFP and CYFP complementation and high fluorescence only if protein A forms an oligomer. As expected, we do observe that proteins that are known to form homo-oligomers show high fluorescence along the diagonal.

Reviewer #2:

Referee Report on "Conserved Interfaces Mediate Multiple Protein-Protein Interactions in a Prokaryotic Metabolon"

The manuscript presents an extensive mapping of protein-protein interactions (PPI) in the E. coli 1-carbon metabolism pathway, using a combination of high-throughput fluorescence assays, mutagenesis, and computational modeling. While the dataset is impressive and the findings intriguing, the biological significance of the reported interactions remains unclear. Below, I outline specific concerns that should be addressed to strengthen the manuscript.

Major Concerns:

1. Limited Biological Interpretation of Intra-Pathway Interaction Enhancement

The authors report a statistically significant enhancement of intra-pathway interactions (Fig. 3E), but the effect size appears modest. It is unclear whether this enhancement is functionally meaningful or merely a byproduct of the experimental method. Could the authors provide additional evidence or theoretical justification for why this level of enhancement would impact metabolic flux or enzyme function?

We thank the referee for the comment. To address this criticism, we performed coarse-grained diffusion-reaction simulations to probe the impact of the folate pathway PPI on metabolic fluxes (see section “*Folate-pathway metabolons show a dramatic increase in metabolic flux for the pathway compared to monomeric enzymes*” in the revised manuscript, lines 428-476). Our model accounts for the protein-protein interaction (based on the experimental MFI values) with patchy particle enzymes which have two interaction patches, one that participates in protein-protein interactions and the second that accounts for substrate binding. Strikingly, our results suggest that incorporation of folate-pathway PPIs results in formation of enzyme clusters which in turn leads to a dramatic speedup of the reaction fluxes in a linear 10-reaction pathway. This speedup for the later reactions in the pathway in the presence of PPIs is about two orders of magnitude compared with that of the bulk reaction scenario in the absence of PPIs.

These results reiterate our earlier findings (Ranganathan, Liu and Shakhnovich, Biophys J. (2023) 122(23):4555-4566) which suggest that clustering of enzymes into metabolons via PPIs can result in a speedup in reaction fluxes in long pathways. In the current paper, we extend the findings of the previous model to incorporate features of PPIs in the folate

pathway accounting for a shared interaction patch for PPIs and a heterogenous interaction map corresponding to the folate pathway. These results are now introduced in the revised manuscript (lines 428-476 and a new Figure 8 is added to present them along with extended view figures EV15, EV16 and EV17). The Discussion section has also been modified (lines 569-586).

2. Translation of MFI Values into Biophysical Parameters

The manuscript heavily relies on mean fluorescence intensity (MFI) values to quantify interactions. However, MFI is not a direct biophysical measurement, making it difficult to interpret the binding dynamics. The authors should attempt to translate these values into more conventional biophysical terms, such as dissociation constants (K_d) or, ideally, the fraction of time each enzyme spends bound to its interaction partner(s). Without this, it is difficult to assess whether these interactions are transient or functionally relevant.

While deducing K_D values based on the fluorescence values is out of scope for this work since we do not have absolute expression levels for any of the proteins, we do have characterized some of the PPI reported in this work using purified proteins in an earlier manuscript (Bhattacharyya *et al*, *Elife*. 2016 Dec 10; 5: e20309). Here we reported K_D values around $3\mu\text{M}$ for interaction between *E.coli* DHFR and PurH, while $8\mu\text{M}$ for DHFR and GlyA using surface plasmon resonance. Considering that DHFR-GlyA (MFI:1002) and DHFR:PurH (MFI:917) are some of the strong interactions reported in our dataset, it is reasonable to assume that most of the strong PPIs have K_D s in the low μM range, while weaker ones have K_D s in the higher μM range. The same range of interactions strengths are reported here for the Fola-Folk pair of purified proteins from *in vitro* Raman spectroscopy study (Fig.4).

Based on the experimentally obtained PPI network and this assumption of K_D ranges, we set up Langevin dynamics simulations with patchy particle enzymes and a diffusion-reaction protocol. Our simulations confirm that enzyme clusters do form in these range of K_D values. Strikingly, the reaction fluxes for the pathway were observed to be several orders of magnitude higher for the clustered state as compared to the bulk reactions suggesting a strong biological significance. These new findings are presented in Fig.8 of the revised manuscript.

3. Lack of Biological Rationale for Conserved Interaction Surfaces

A key observation is that proteins in the metabolon use a common interface to interact with all their partners, resulting in exclusively dimeric interactions. While this is an interesting structural feature, its biological significance is not well explained. Why would evolution favor this particular interaction mode? Would it constrain or facilitate dynamic assembly and disassembly of the metabolon? A mechanistic model explaining the functional consequences of this interaction topology would strengthen the study.

We thank the referee for the comment. To address this question, we performed simulations with patchy particle enzymes where the interaction between different enzymes is facilitated via a shared patch. In our simulations, for relatively weak K_D values ($10\mu\text{M}$ to 100mM) we observe that enzyme clusters do form, and result in increased metabolic fluxes for the pathway within metabolons as compared to reactions that occur in the bulk.

A protein-protein interaction surface that is distinct to the substrate binding face of the enzyme could allow for a modular evolutionary design which decouples substrate binding and PPIs (which facilitate metabolon formation). A defined patch also allows greater specificity with respect to interaction partners and tight stoichiometric control over enzymes that can be part of the metabolon. Use of singlish hubs guarantees that one interaction must dissociate before engaging in another one. Shared interfaces might also pose a natural limit on the size of the metabolon, which otherwise can form giant assemblies that completely sequester proteins.

Whether this is an optimal regime for the metabolons to be effective promoters of reaction fluxes would, however, require a detailed study probing various interaction patch geometries and interaction strengths. This will indeed be a subject of our detailed future theoretical study.

Minor Issues:

4. Comparisons with Other Metabolons

The authors suggest that the observed interaction architecture might be a general feature of metabolons. Providing a comparison with known metabolon structures (e.g., purinosomes) would help contextualize these findings.

Whether the findings of our study are a general feature of metabolons would need more investigation. Unfortunately, structural information about purinosomes is not available. Mapping of PPI using a luciferase-based Tango reporter system showed that three enzymes hPPAT, hTrifGART, and hFGAMS of the purine biosynthesis pathway form a core structure, while other proteins associate on to them. However, it is unknown if the enzymes use a common interface or can associate simultaneously. In the only other study where some degree of structural information is available for metabolons, cross-linking followed by mass spectrometry showed that citrate synthase of TCA cycle uses two distinct PPI interfaces to interact with malate dehydrogenase and aconitase, akin to FOLD in our study. Like what we found in this study, the PPI interface was distinct from the active site. This information is now included in the Discussion section (lines 531-541) of the revised manuscript.

In conclusion, while the manuscript presents a valuable dataset and some novel insights, the biological significance of the findings is not sufficiently clear. Addressing the concerns outlined above, particularly by providing a more quantitative analysis of binding strengths and a mechanistic rationale for the observed interaction architecture, would greatly improve the impact of the study.

We hope that the new Figure 8 and our responses to the above-mentioned criticisms will address the biological significance of our findings.

Reviewer #3:

The authors utilized biomolecular fluorescence complementation (BiFc) to extensively map weak protein-protein interactions (PPIs) in the E.coli 1-carbon metabolism pathway. Thirty-five

enzymes are fused to the N- and C-YFP, and the complete 1225 pairs are analyzed using flow cytometry. Discovered interactions are further assessed using various techniques. They demonstrated certain conserved region interacts with multiple partners. Though the systematical map of PPIs using BiFc may have some background noise from non-specific interactions, it provides a high throughput approach to uncover potential PPIs among intra- and inter- pathway enzymes. Researchers who interest in PPIs among multiple proteins may find this study informative.

Major points

1. The authors describe, "for each pair, fluorescence was measured in two configurations: NYFP-X/CYFP-Y and NYFP-Y/CYFP-X, where X and Y are the two proteins in question" and all the pairs were experimentally analyzed. However, "Fig 2A was generated by selecting the highest fluorescence value obtained in any of the configuration". It's known and understandable that the two configurations could generate different fluorescence values. But it's misleading to just show the highest value. The authors need to present the entire dataset from the two configurations in the matrix and use that to re-do the analysis presented in Figure 2E.

We thank the reviewer for the comment. We have now re-analyzed the asymmetric matrix (taking data from both configurations) to generate a figure analogous to Fig 2E. This new analysis is presented in Fig EV1B. As expected, the broad conclusions about cluster formation are still similar to Fig 2E and the clusters still remain highly statistically significant, however in some cases, the mean fluorescence of the cluster is lower. This is rationalizable, since the interaction may not be optimal in a one configuration. The complete matrix has been included in Table EV2, and a color-coded matrix analogous to Fig 2A is now shown in Fig EV1A.

2. The authors claim that "we found that the increase in inducer concentration causes a sharp increase in YFP fluorescence (Fig 2B, C and D), indicating that the observed fluorescence is not non-specific, rather it arises from specific protein-protein interactions that are strongly concentration dependent." Both non-specific and specific PPIs can lead to YFP reconstitution, the protein expression dependent of YFP fluorescence cannot prove the interactions are specific.

We agree with the reviewer's comment. The data showing increase in fluorescence with inducer concentration is important as it helped us decide which range of inducers gave us the highest signal to noise ratio. However, it is true that it does not specifically signify whether the interactions are specific or non-specific and we have now removed this claim from the manuscript.

3. The authors say, "Further the resultant spectrum was not the mathematical sum of individual spectra obtained with DHFR and FolK before interaction." However, the mathematical sum spectrum of DHFR and FolK is missing in the Figure 5A. The solely appearance of "new features around 1663 cm^{-1} " is not sufficient to convince others.

We thank the reviewer for this insightful comment. Earlier we did not explicitly include the mathematical sum of the individual DHFR and FolK (HPPK) spectra in Figure 5A. In the revised manuscript, we have included the spectral comparisons in Figure EV9.

As noted in the text, the resultant spectrum deviates from what would be anticipated if the proteins were non-interacting—particularly evident through the emergence of a distinct new peak around 1663 cm⁻¹, which is absent in both individual spectra. This non-additivity is a hallmark of conformational changes due to complex formation. To further substantiate this claim, we included appropriate negative controls (Figure EV9B and C, DHFR–ADK and FolK–ADK), where no such deviation or new spectral feature was observed; the spectra in these cases matched the expected mathematical sums, supporting the specificity of the DHFR–FolK interaction.

Additionally, the Principal Component Analysis (PCA) shown in Figure 5G provides an orthogonal and unsupervised validation of this interaction. Here, DHFR variants that are capable of interacting with FolK (WT, D132H, D144N, N23V) form a distinct spectral cluster, whereas non-interacting mutants (e.g., D131N, F140S, R159L) are clearly separated. This global spectral divergence, independent of any one feature, strongly supports the structural uniqueness of the interacting complex and the interpretation that the spectral profile is not merely additive.

This is now included in Figure EV9 and lines 293-298 and lines 320-344 of the revised manuscript.

4. AF3 multimer modeling rely on a random seed; it is possible that different seeds will yield different predicted structures, yielding an ensemble of structure rather than a unique prediction. Is this the case for the AF models studied in this manuscript? In other words, how representative are the structures used for metadynamics simulations with respect to the ensemble of AF3 predictions?

The metadynamics simulations were run along the distance between the center of mass of the two chains as the reaction coordinate. This ensures that the system is not strictly biased by the initial binding configuration. Rather, the simulation protocol allows for unbinding and rebinding events making it agnostic to the initial AF3 configuration.

Minor point

1. The split point for YFP used in this study is at 155/156, which has been shown with moderate self-assemble probability. Reference "Ohashi, K., Kiuchi, T., Shoji, K., Sampei, K., & Mizuno, K. (2012). Visualization of cofilin-actin and Ras-Raf interactions by bimolecular fluorescence complementation assays using a new pair of split Venus fragments. *Biotechniques*, 52(1), 45-50." Just want to bring this to authors attention and I don't expect any experiment about this matter.

Thank you for this comment. We have now included this information along with the reference on lines 130-131.

22nd Jul 2025

Manuscript Number: MSB-2025-12868R

Title: Conserved interfaces mediate multiple protein-protein interactions in a prokaryotic metabolon

Dear Prof Shakhnovich,

Thank you for the submission of your revised manuscript to Molecular Systems Biology. We have now received the enclosed reports from the referees that were asked to re-assess it. As you will see the reviewers are now globally supportive and I am pleased to inform you that we will be able to accept your manuscript pending the following final amendments:

- 1) In the main manuscript file, please include keywords to max. 5.
- 2) Please remove the "Significance statement" from the manuscript. This paragraph may be useful though for preparing the Synopsis text we request below.
- 3) Please rename "Declaration" to "Disclosure and competing interests statement". We updated our journal's competing interests policy in January 2022 and request authors to consider both actual and perceived competing interests. Please review the policy <https://www.embopress.org/competing-interests> and update your competing interests if necessary.
- 4) Author contributions: Please remove it from the manuscript and specify author contributions in our submission system. CRedit has replaced the traditional author contributions section because it offers a systematic machine-readable author contributions format that allows for more effective research assessment. You are encouraged to use the free text boxes beneath each contributing author's name to add specific details on the author's contribution. More information is available in our guide to authors:
<https://www.embopress.org/page/journal/17574684/authorguide#authorshipguidelines>
- 5) References: Please correct the reference citation in the reference list to be alphabetical (not numerical). Where there are more than 10 authors on a paper, only the first 10 should be listed, followed by "et al.". Please check "Author Guidelines" for more information.
<https://www.embopress.org/page/journal/17574684/authorguide#referencesformat>
- 6) Data not shown: We do not allow statements/conclusions with "data not shown". All data referred to in the paper should be displayed in the main or Expanded View figures. Please remove from page 22.
- 7) In the Methods, please take care of the following:
 - Please ensure that a statement on whether or not blinding was done is included in the Methods even if no blinding was done. Please also be sure to update the Author Checklist with this information and where it can be found in the manuscript.
- 8) Please place individual sections of the manuscript in the following order: Title page - Abstract & Keywords - Introduction - Results - Discussion - Methods - Data Availability - Acknowledgements - Disclosure and Competing Interests Statement - References - Figure Legends - Expanded View Figure Legends.
- 9) For the figures and figure legends, please take care of the following:
 - There is a callout for Fig S12, but no such figure is uploaded to your submission.
 - Please define the annotated p values ****/***/**/* as well as provide the exact p-values for the same in the legend of figure EV2 B, D as appropriate.
 - Please note that the exact p values are not provided in the legends of figures 2E, 3A, B; EV1 B,
 - Please indicate the statistical test used for data analysis in the legends of figures 2E, 3A, B; 4D, E; EV1 B, EV2 D
 - Please note that the box plots need to be defined in terms of minima, maxima in the legends of figures 2E, 3A-C; EV1 B
 - Please note that the box plots need to be defined in terms of minima, maxima, centre, bounds of box and whiskers, and percentile in the legends of figure EV2 D
 - Please note that information related to n is missing in the legends of figures 3C, 4D, E
 - Please note that the error bars are not defined in the legend of figure EV2 B
 - Please note that scale bar and its definition are missing for figure EV4.
 - Typically we do not have more than 10 EV figures for a manuscript. As you have 17 EV figures, we would encourage you to compile some of these figures into an Appendix file, with the legends under each figure with the file names and callouts renamed as Appendix Figure SX. The appendix should be uploaded in PDF format and needs a table of contents with page numbers and the title "Appendix for [manuscript title]"
- 10) Tables: Please rename Tables EV1-EV4 to Dataset EV1-EV4. Each dataset will need its legend removed from the manuscript and added to the corresponding file in a separate tab. Please also update their callouts in main manuscript text and the name of the files in your submission and the titles of the files in their respective legends.
- 11) Synopsis:
 - Synopsis image: Please provide a graphic that summarises the main findings of the manuscript on a glance and upload it as a high-resolution jpeg file 550 pixels wide x (300-600) pixels high.
 - Synopsis text: Please provide a short standfirst (maximum of 300 characters, including space), limit the bullet points to max. 5 and upload it as a separate .doc file. Please write the bullet points to summarise the key NEW findings. They should be designed to be complementary to the abstract - i.e. not repeat the same text. We encourage inclusion of key acronyms and quantitative information (maximum of 30 words / bullet point).

12) As part of the EMBO Publications transparent editorial process initiative (see our policy here: https://www.embopress.org/transparent-process#Review_Process), Molecular Systems Biology will publish online a Peer Review File (PRF) to accompany accepted manuscripts. This file will be published in conjunction with your paper and will include the anonymous referee reports, your point-by-point response and all pertinent correspondence relating to the manuscript. Let us know whether you agree with the publication of the PRF and as here, if you want to remove or not any figures from it prior to publication. Please note that the Authors checklist will be published at the end of the PRF.

13) After your paper is published, we may promote it on social media. If you have any handles or hashtags for Bluesky you would like included, please let us know.

14) Please provide a point-by-point letter INCLUDING my comments and your detailed responses (as Word file).

I look forward to reading a new revised version of your manuscript as soon as possible.

Yours sincerely,

Poonam Bheda, PhD
Scientific Editor
Molecular Systems Biology

Reviewer #1:

The authors have carefully and convincingly addressed all reviewers' comments. The manuscript has therefore further improved. Very good work

Reviewer #2:

The resubmitted manuscript fully meets the concerns I raised. The authors now demonstrate functional relevance by adding diffusion-reaction simulations that show a ~100-fold flux boost, calibrate MFI signals with published and new K_d measurements to anchor binding strengths, provide an evolutionary rationale for the single shared interface supported by additional modeling, and situate their findings within other metabolon examples such as the TCA cycle and the purinosome. These additions are clear, quantitative, and directly responsive, and I therefore recommend the paper for acceptance.

Reviewer #3:

The revision has addressed all my concerns.

Minor point:

In Figure EV9, HPPK and Folk are both used. Please ensure consistent nomenclature to avoid confusion.

1) In the main manuscript file, please include keywords to max. 5.

5 keywords have been included.

2) Please remove the "Significance statement" from the manuscript. This paragraph may be useful though for preparing the Synopsis text we request below.

This has been removed.

3) Please rename "Declaration" to "Disclosure and competing interests statement". We updated our journal's competing interests policy in January 2022 and request authors to consider both actual and perceived competing interests. Please review the policy <https://www.embopress.org/competing-interests> and update your competing interests if necessary.

We have reviewed this and changed "Declaration" to "Disclosure and competing interests statement".

4) Author contributions: Please remove it from the manuscript and specify author contributions in our submission system. CRediT has replaced the traditional author contributions section because it offers a systematic machine-readable author contributions format that allows for more effective research assessment. You are encouraged to use the free text boxes beneath each contributing author's name to add specific details on the author's contribution. More information is available in our guide to authors:

<https://www.embopress.org/page/journal/17574684/authorguide#authorshipguidelines>

Author contributions have been removed from the manuscript.

5) References: Please correct the reference citation in the reference list to be alphabetical (not numerical). Where there are more than 10 authors on a paper, only the first 10 should be listed, followed by "et al.". Please check "Author Guidelines" for more information.

<https://www.embopress.org/page/journal/17574684/authorguide#referencesformat>

The reference section has been updated with the MSB style.

6) Data not shown: We do not allow statements/conclusions with "data not shown". All data referred to in the paper should be displayed in the main or Expanded View figures. Please remove from page 22.

This statement has now been removed.

7) In the Methods, please take care of the following:

- Please ensure that a statement on whether or not blinding was done is included in the Methods even if no blinding was done. Please also be sure to update the Author Checklist with this information and where it can be found in the manuscript.

A statement about blinding has been included in the methods section and the Author Checklist.

8) Please place individual sections of the manuscript in the following order: Title page - Abstract & Keywords - Introduction - Results - Discussion - Methods - Data Availability - Acknowledgements - Disclosure and Competing Interests Statement - References - Figure Legends - Expanded View Figure Legends.

The manuscript has been formatted in this order.

9) For the figures and figure legends, please take care of the following:
- There is a callout for Fig S12, but no such figure is uploaded to your submission.

This was by mistake, and the name has been changed appropriately.

- Please define the annotated p values ****/****/**/* as well as provide the exact p-values for the same in the legend of figure EV2 B, D as appropriate.

The * definition as well as the exact p-values have now been included.

- Please note that the exact p values are not provided in the legends of figures 2E, 3A, B; EV1 B

The exact p-values have now been included.

- Please indicate the statistical test used for data analysis in the legends of figures 2E, 3A, B; 4D, E; EV1 B, EV2 D

The statistical test has now been included in the legend.

- Please note that the box plots need to be defined in terms of minima, maxima in the legends of figures 2E, 3A-C; EV1 B

This is now defined in the legends.

- Please note that the box plots need to be defined in terms of minima, maxima, centre, bounds of box and whiskers, and percentile in the legends of figure EV2 D

This is now defined in the legends.

- Please note that information related to n is missing in the legends of figures 3C, 4D, E

This is now included in the legends.

- Please note that the error bars are not defined in the legend of figure EV2 B

The error bar is now defined in the legends.

- Please note that scale bar and its definition are missing for figure EV4.

The scale bar is now included in the figure.

- Typically we do not have more than 10 EV figures for a manuscript. As you have 17 EV figures, we would encourage you to compile some of these figures into an Appendix file, with the legends under each figure with the file names and callouts renamed as Appendix Figure SX. The appendix should be uploaded in PDF format and needs a table of contents with page numbers and the title "Appendix for [manuscript title]"

We have moved 7 EV figures to the appendix now, which is a separate pdf. The EV figures have accordingly been renamed.

10) Tables: Please rename Tables EV1-EV4 to Dataset EV1-EV4. Each dataset will need its legend removed from the manuscript and added to the corresponding file in a separate tab. Please also update their callouts in main manuscript text and the name of the files in your submission and the titles of the files in their respective legends.

The filenames and their callouts have been renamed.

11) Synopsis:

- Synopsis image: Please provide a graphic that summarises the main findings of the manuscript on a glance and upload it as a high-resolution jpeg file 550 pixels wide x (300-600) pixels high.

A synopsis image has been uploaded.

- Synopsis text: Please provide a short standfirst (maximum of 300 characters, including space), limit the bullet points to max. 5 and upload it as a separate .doc file. Please write the bullet points to summarise the key NEW findings. They should be designed to be complementary to the abstract - i.e. not repeat the same text. We encourage inclusion of key acronyms and quantitative information (maximum of 30 words / bullet point).

A synopsis text along with the blurb has been uploaded as a separate doc file.

12) As part of the EMBO Publications transparent editorial process initiative (see our policy here: https://www.embopress.org/transparent-process#Review_Process), Molecular Systems Biology will publish online a Peer Review File (PRF) to accompany accepted manuscripts. This file will be published in conjunction with your paper and will include the anonymous referee reports, your point-by-point response and all pertinent correspondence relating to the manuscript. Let us know whether you agree with the publication of the PRF and as here, if you want to remove or not any figures from it prior to publication. Please note that the Authors checklist will be published at the end of the PRF.

We agree with the publication of the peer review file (PRF).

13) After your paper is published, we may promote it on social media. If you have any handles or hashtags for Bluesky you would like included, please let us know.

None.

14) Please provide a point-by-point letter INCLUDING my comments and your detailed responses (as Word file).

A response letter detailing this has been attached.

14th Aug 2025

Manuscript number: MSB-2025-12868RR

Title: Conserved interfaces mediate multiple protein-protein interactions in a prokaryotic metabolon

Dear Prof Shakhnovich,

Congratulations on an excellent manuscript, I am pleased to inform you that your manuscript has been accepted for publication in Molecular Systems Biology. Thank you for your comprehensive response to referee concerns. It has been a pleasure to work with you to get this to the acceptance stage.

Yours sincerely,

Poonam Bheda, PhD
Scientific Editor
Molecular Systems Biology
